# Mean-Field Analysis for Learning Subspace-Sparse Polynomials with Gaussian Input

**Ziang Chen**
Department of Mathematics
Massachusetts Institute of Technology
Cambridge, MA 02139
ziang@mit.edu

**Rong Ge**
Department of Computer Science and Department of Mathematics
Duke University
Durham, NC 27708
rongge@cs.duke.edu

## Abstract

In this work, we study the mean-field flow for learning subspace-sparse polynomials using stochastic gradient descent and two-layer neural networks, where the input distribution is standard Gaussian and the output only depends on the projection of the input onto a low-dimensional subspace. We establish a necessary condition for SGD-learnability, involving both the characteristics of the target function and the expressiveness of the activation function. In addition, we prove that the condition is almost sufficient, in the sense that a condition slightly stronger than the necessary condition can guarantee the exponential decay of the loss functional to zero.

## 1 Introduction

Neural Networks (NNs) are powerful in practice to approximate mappings on certain data structures, such as Convectional Neural Networks (CNNs) for image data, Graph Neural Networks (GNNs) for graph data, and Recurrent Neural Networks (RNNs) for sequential data, stimulating numerous breakthroughs in application of machine learning in many branches of science, engineering, etc. The surprising performance of neural networks is often explained by arguing that neural networks automatically learns useful representations of the data. However, how simple training procedures such as stochastic gradient descent (SGD) extract features remains a major open problem.

Optimization of neural networks has received lots of attention. For simpler networks such as linear neural networks, local minima are also globally optimal [14, 15, 17]. However, this is not true for nonlinear networks even of depth 2 [23]. Neural Tangent Kernel (NTK, [4, 12, 13]) is a line of work that establishes strong convergence results for wide neural networks. However, in the NTK regime, neural network is equivalent to a kernel, which cannot learn useful features based on the target function. Such limitation prevents neural networks in NTK regime from efficiently learning even simple single index models [28].

As an alternative, the behavior of SGD can also be understood via mean-field analysis, for both two-layer neural networks [9, 19, 20, 22, 24, 25] and multi-layer neural networks [5, 21, 22]. Neural networks in the mean-field regime have the potential to do feature learning. Recently, [1] showed an interesting setup where a two-layer neural network can learn representations if the target function satisfies a merged-staircase property. More precisely, [1] considers a sparse polynomial as a

polynomial $f^* : \mathbb{R}^d \to \mathbb{R}$ defined on the hypercube $\{-1, 1\}^d$, i.e., $f^*(x) = h^*(z) = h^*(x_I)$ where $z = x_I = (x_i)_{i \in I}$, $I$ is an unknown subset of $\{1, 2, \ldots, d\}$ with $|I| = p$, and $h^* : \{-1, 1\}^p \to \mathbb{R}$ is a function on the subset of coordinates in $I$. They prove that a condition called the merged-staircase property is necessary and in some sense sufficient for learning such $f^*$ using SGD and two-layer neural networks. The merged-staircase property proposed in [1] states that all monomials of $h^*$ can be ordered such that each monomial contains at most one $z_i$ that does not appear in any previous monomial. For example, $h^*(z) = z_1 + z_1 z_2 + z_1 z_2 z_3$ satisfies the merged-staircase property while $h^*(z) = z_1 + z_1 z_2 z_3$ does not. Results on similar structures can also be found in [3]. The work [2] proposes the concept of leap complexity and generalizes the results in [1] to a larger family of sparse polynomials.

In this work, we consider "subspace-sparse" polynomial that is more general. Concretely, let $f^*(x) = h^*(z) = h^*(x_V)$, where $V$ is a subspace of $\mathbb{R}^d$ with $\dim(V) = p \ll d$, $x_V$ is the orthogonal projection of $x$ onto the subspace $V$, and $h^* : V \to \mathbb{R}$ is an underlying polynomial map. In other words, the sparsity is in the sense that $f^*(x)$ only depends on the projection of the input $x \in \mathbb{R}^d$ in a low-dimensional subspace. Throughout this paper, the input data distribution is the standard $d$-dimensional normal distribution, i.e., $x \sim \mathcal{N}(0, I_d)$, which is rotation-invariant in the sense that $Ox \sim \mathcal{N}(0, I_d)$ for any orthogonal matrix $O \in \mathbb{R}^{d \times d}$. Similar rotation-invariant/basis-free settings are also considered in some recent studies, including [2, 8, 10, 11].

**Our contribution and related works**   Our first contribution is a basis-free necessary condition for SGD-learnability. More specially, we propose the reflective property of the underlying polynomial $h^* : V \to \mathbb{R}$ with respect to some subspace $S \subset V$, which also involves the expressiveness of the activation function. We prove that as long as the reflective property is satisfied with respect to nontrivial $S$, the training dynamics cannot learn any information about the behavior of $h^*$ on $S$ (see Theorem 3.4). Therefore the loss functional will be bounded away from 0 during the whole training procedure.

One key point is that our reflective property precisely characterizes the necessary expressiveness of the activation function. If the activation function is expressive enough, the reflective property equivalently recovers a necessary condition characterized by isoLeap [2] that is the maximal leap complexity over all orthonormal basis and can be viewed as a basis-free generalization of the merged-staircase property. This also indicates that our necessary condition is a bit weaker. Other related rotation-invariant conditions in the previous literature include leap exponent/index [8, 10], subspace conditioning [10] and even-symmetric directions [11]. The analysis in [10, 11] is for training the first layer for finitely many iterations with fixed second-layer, and [8] studies the joint learning dynamics where they assume that for any fixed first layer, the optimal parameters in second layer can be found efficiently and reformulate the loss as a function of the first layer. Differently and more generally, our analysis for the necessary condition does not require specific learning strategies and works for any learning rates satisfying some mild conditions.

Our second contribution is a sufficient condition for SGD-learnability that is also basis-free and is slightly stronger than the necessary condition. In particular, we show that if the training dynamics cannot be trapped in any proper subspace of $V$, then one can choose the initial parameter distribution and the learning rate such that the loss functional decays to zero exponentially fast with dimension-free rates (see Theorem 4.3). Our training strategy is inspired by [1] with the difference that we take the average of $p$ independent training trajectories, which can lift some linear independence property required for polynomials on hypercube to algebraic independence in the general polynomial setting.

**Technical challenges**   It may seem simple to leave the standard basis and generalize the results of [1, 2] to learn subspaces, because SGD itself is independent of the basis, and we can consider a symmetric Gaussian input distribution. However, there are some significant barriers that motivated our training process. The condition and the analysis in [1, 2] rely on an orthonormal basis of the input space $\mathbb{R}^d$. This is natural for polynomials on the hypercube $\{-1, 1\}^d$, but not for general polynomials on $\mathbb{R}^d$. Particularly, their theory does not work for Gaussian input data $x \sim \mathcal{N}(0, I_d)$, which is probably the most common distribution in data science, unless an orthonormal basis of $\mathbb{R}^d$ is specified and $V$ is known to be spanned by $p$ elements in the basis. In this work, we consider a more general setting in which specifying a basis is not required and the space $V$ can be any $p$-dimensional subspace of $\mathbb{R}^d$. This setting is consistent with the rotation-invariant property of $\mathcal{N}(0, I_d)$ and introduces more difficulties since less knowledge of $V$ is available prior to training.

**Organization** The rest of this paper will be organized as follows. We introduce some preliminaries on mean-field dynamics in Section 2. The basis-free necessary and sufficient conditions for SGD-learnability are discussed in Section 3 and Section 4, respectively. We conclude in Section 5.

## 2  Preliminaries on Mean-Field Dynamics

The mean-field dynamics describes the limiting behavior of the training procedure when the step-size/learning rate converges to zero, i.e., the evolution of a neuron converges to the solution of a differential equation with continuous time, and when the number of neurons converges to infinity, i.e., the empirical distribution of all neurons converges to some limiting probability distribution. For two-layer neural networks, some quantitative results are established in [20] that characterize the distance between the SGD trajectory and the mean-field evolution flow, and these results are further improved as dimension-free in [19]. Such results suggest that analyzing the mean-field flow is sufficient for understanding the SGD trajectory in some settings. In this section, we briefly review the setup of two-layer neural networks, SGD, and their mean-field versions, following [19, 20].

**Two-layer neural network and SGD** The two-layer neural network is of the following form:

$$f_{\mathrm{NN}}(x; \Theta) := \frac{1}{N} \sum_{i=1}^{N} \tau(x; \theta_i) = \frac{1}{N} \sum_{i=1}^{N} a_i \sigma(w_i^\top x), \tag{2.1}$$

where $N$ is the number of neurons, $\Theta = (\theta_1, \theta_2, \ldots, \theta_N)$ with $\theta_i = (a_i, w_i) \in \mathbb{R}^{d+1}$ is the set of parameters, and $\sigma : \mathbb{R} \to \mathbb{R}$ is the activation functions with $\tau(x; \theta) := a\sigma(w^\top x)$ for $\theta = (a, w)$. Then the task is to find some parameter $\Theta$ such that the $\ell_2$-distance between $f^*$ and $f_{\mathrm{NN}}$ is minimized:

$$\min_{\Theta} \ \mathcal{E}_N(\Theta) := \frac{1}{2} \mathbb{E}_{x \sim \mathcal{N}(0, I_d)} \left[ |f^*(x) - f_{\mathrm{NN}}(x; \Theta)|^2 \right]. \tag{2.2}$$

In practice, a widely used algorithm for solving (2.2) is the stochastic gradient descent (SGD) that iterates as

$$\theta_i^{(k+1)} = \theta_i^{(k)} + \gamma^{(k)} \left( f^*(x_k) - f_{\mathrm{NN}}(x_k; \Theta^{(k)}) \right) \nabla_\theta \tau(x_k; \theta_i^{(k)}), \tag{2.3}$$

where $x_k$, $k = 1, 2, \ldots$ are the i.i.d. samples drawn from $\mathcal{N}(0, I_d)$ and $\gamma^{(k)} = \mathrm{diag}(\gamma_a^{(k)}, \gamma_w^{(k)} I_d) \succeq 0$ is the stepsize or the learning rate. In this paper, we only consider the one-pass model with each data point being used exactly once, following [19].

**Mean-field dynamics** One can generalize (2.1) to an infinite-width two-layer neural network:

$$f_{\mathrm{NN}}(x; \rho) := \int \tau(x; \theta) \rho(d\theta) = \int a\sigma(w^\top x) \rho(da, dw),$$

where $\rho \in \mathcal{P}(\mathbb{R}^{d+1})$ is a probability measure on the parameter space $\mathbb{R}^{d+1}$, and generalize the loss/energy functional (2.2) to

$$\mathcal{E}(\rho) := \frac{1}{2} \mathbb{E}_{x \sim \mathcal{N}(0, I_d)} \left[ |f^*(x) - f_{\mathrm{NN}}(x; \rho)|^2 \right].$$

We will use $\mathcal{P}(X)$ to denote the collection of probability measures on a space $X$ throughout this paper. The limiting behavior of the SGD trajectory (2.3) when $\gamma^{(k)} \to 0$ and $N \to \infty$ can be described by the following mean-field dynamics:

$$\begin{cases} \partial_t \rho_t = \nabla_\theta \cdot (\rho_t \xi(t) \nabla_\theta \Phi(\theta; \rho_t)), \\ \rho_t \big|_{t=0} = \rho_0, \end{cases} \tag{2.4}$$

where $\xi(t) = \mathrm{diag}(\xi_a(t), \xi_w(t) I_d) \in \mathbb{R}^{(d+1) \times (d+1)}$ with $\xi_a(t) \geq 0$ and $\xi_w(t) \geq 0$ being the learning rates and

$$\Phi(\theta; \rho) = a \mathbb{E}_{x \sim \mathcal{N}(0, I_d)} \left[ (f_{\mathrm{NN}}(x; \rho) - f^*(x)) \sigma(w^\top x) \right].$$

One can also write $\Phi(\theta; \rho)$ as

$$\Phi(\theta; \rho) = V(\theta) + \int U(\theta, \theta') \rho(d\theta'),$$

where

$$V(\theta) = -a\mathbb{E}_x\left[f^*(x)\sigma(w^\top x)\right] \quad \text{and} \quad U(\theta, \theta') = aa'\mathbb{E}_x\left[\sigma(w^\top x)\sigma((w')^\top x)\right]. \tag{2.5}$$

The PDE (2.4) is understood in the weak sense, i.e., $\rho_t$ is a solution to (2.4) if and only if $\rho_t\big|_{t=0} = \rho_0$ and

$$\iint \left(-\partial_t \eta + \nabla_\theta \eta \cdot (\xi(t)\nabla_\theta \Phi(\theta; \rho_t))\right)\rho_t(d\theta)dt = 0, \quad \forall \eta \in \mathcal{C}_c^\infty(\mathbb{R}^{d+1} \times (0, +\infty)),$$

where $\mathcal{C}_c^\infty(\mathbb{R}^{d+1} \times (0, +\infty))$ is the collection of all smooth and compactly supported functions on $\mathbb{R}^{d+1} \times (0, +\infty)$. It can also be computed that the energy functional is non-increasing along $\rho_t$:

$$\frac{d}{dt}\mathcal{E}(\rho_t) = -\int \nabla_\theta \Phi(\theta; \rho_t)^\top \xi(t)\nabla_\theta \Phi(\theta; \rho_t)\rho_t(d\theta) \leq 0. \tag{2.6}$$

There have been standard results in the existing literature that provide dimension-free bounds for the distance between the empirical distribution of the parameters generalized by (2.3) and the solution to (2.4). For the simplicity of reading, we will not present those results and the proof; interested readers are referred to [19]. In the rest of this paper, we will focus on the analysis of (2.4) and briefly discuss the sample complexity results implied by our mean-field analysis.

# 3   Necessary Condition for SGD-Learnability

This section introduces a condition that can prevent SGD from recovering all information about $f^*$, or in other words, prevent the loss functional $\mathcal{E}(\rho_t)$ decaying to a value sufficiently close to 0.

## 3.1   Reflective Property

Before rigorously presenting our main theorem, we state the assumptions used in this section.

**Assumption 3.1.** *Assume that the followings hold:*

  (i) *The activation function $\sigma : \mathbb{R} \to \mathbb{R}$ is twice continuously differentiable with $\|\sigma\|_{L^\infty(\mathbb{R})} \leq K_\sigma$, $\|\sigma'\|_{L^\infty(\mathbb{R})} \leq K_\sigma$, and $\|\sigma''\|_{L^\infty(\mathbb{R})} \leq K_\sigma$ for some constant $K_\sigma > 0$.*

  (ii) *The learning rates $\xi_a, \xi_w : \mathbb{R}_{\geq 0} \to \mathbb{R}$ satisfy that $\|\xi_a\|_{L^\infty(\mathbb{R}_{\geq 0})} \leq K_\xi$ and $\|\xi_w\|_{L^\infty(\mathbb{R}_{\geq 0})} \leq K_\xi$ for some constant $K_\xi > 0$. Furthermore, $\xi_a$ and $\xi_w$ are Lipschitz continuous with $\int_0^{+\infty} \xi_a(t)dt = +\infty$ and $\int_0^{+\infty} \xi_w(t)dt = +\infty$.*

  (iii) *The initialization is $\rho_0 = \rho_a \times \rho_w$ such that $\rho_a$ is symmetric and is supported in $[-K_\rho, K_\rho]$ for some constant $K_\rho > 0$.*

In Assumption 3.1, the Condition (i) is satisfied by some commonly used activation functions, such as $\sigma(x) = \frac{1}{1+e^{-x}}$ and $\sigma(x) = \cos(x)$, and is required for establishing the existence and uniqueness of the solution to (2.4). The Condition (ii) and (iii) are also standard and easy to satisfy in practice.

**Remark 3.2.** *The symmetry of $\rho_a$ implies that $f_{NN}(x; \rho_0) = 0$. Therefore, the initial loss $\mathcal{E}(\rho_0) = \frac{1}{2}\mathbb{E}_x[|f^*(x)|^2] = \frac{1}{2}\mathbb{E}_{x_V}[|h^*(x_V)|^2] = \frac{1}{2}\mathbb{E}_z[|h^*(z)|^2]$, where $x_V = z \sim \mathcal{N}(0, I_V)$, can be viewed as a constant depending only on $h^*$ and $p$, independent of $d$. Noticing also the decay property (2.6), the loss at any time $t$ can be bounded as $\mathcal{E}(\rho_t) \leq \mathcal{E}(\rho_0) = \frac{1}{2}\mathbb{E}_z[|h^*(z)|^2]$.*

The main goal of this section is to generalize the merged-staircase property in a basis-free setting for general polynomials. Without a standard basis, it is hard to talk about having a "staircase" of monomials. Even with a fixed basis, it is still nontrial to define the merged-staircase property for general polynomials since the analysis in [1] highly depends on $z_i^2 = 1$ that is only true for polynomials on the hypercube. Instead, we use the observation that when a function does not satisfy the merged staircase property, it implies that two of the variables will behave the same in the training dynamics. Such a symmetry can be generalized to the basis-free setting for general polynomials and we summarize this as the following reflective property:

**Definition 3.3** (Reflective property). *Let $S \subset V \subset \mathbb{R}^d$ be a subspace of $V$. We say that the underlying polynomial $h^* : V \to \mathbb{R}$ satisfies the reflective property with respect to the subspace $S$ and the activation function $\sigma$ if*

$$\mathbb{E}_{z \sim \mathcal{N}(0, I_V)} \left[ h^*(z) \sigma' \left( u + v^\top z_S^\perp \right) z_S \right] = 0, \quad \forall \, u \in \mathbb{R}, \, v \in V, \tag{3.1}$$

*where $z_S = \mathcal{P}_S^V(z)$ and $z_S^\perp = z - \mathcal{P}_S^V(z)$, with $\mathcal{P}_S^V : V \to S$ being the orthogonal projection from $V$ onto $S$.*

The reflective property defined above is closely related to the merged-staircase property in [1]. Let us illustrate the intuition using a simple example. Consider $V = \mathbb{R}^3$ and $h^*(z) = z_1 + z_1 z_2 z_3$. Then $h^*$ does not satisfy the merged-staircase property since $z_1 z_2 z_3$ involves two new coordinates that do not appear in the first monomial $z_1$. In our setting, this $h^*$ satisfies the reflective with respect to $S = \text{span}\{e_2, e_3\}$, where $e_i$ is the vector in $\mathbb{R}^3$ with the $i$-th entry being 1 and other entries being 0. More specifically, for $z = (z_1, z_2, z_3)$, one has that $z_S = (0, z_2, z_3)$ and $z_S^\perp = (z_1, 0, 0)$. Thus, one has for any $u \in \mathbb{R}$ and $v \in V$ that $\sigma' \left( u + v^\top z_S^\perp \right)$ is independent of $z_2, z_3$ and that

$$\mathbb{E}_{z_2, z_3} \left[ h^*(z) \sigma' \left( u + v^\top z_S^\perp \right) z_S \right] = \sigma' \left( u + v^\top z_S^\perp \right) \mathbb{E}_{z_2, z_3} \left[ \left( 0, z_1 z_2 + z_1 z_2^2 z_3, z_1 z_3 + z_1 z_2 z_3^2 \right) \right]$$
$$= (0, 0, 0),$$

which leads to (3.1). One can see from this example that satisfying the reflective property with respect to a nontrivial subspace $S \subset V$ is in the same spirit as not satisfying the merged-staircase property. Furthermore, the reflective property is rotation-invariant, meaning that using a different orthonormal basis does not change the property. In this sense, our proposed condition is more general than that in [1]. We also remark that there have been other rotation-invariant conditions generalizing [1], see e.g., [2, 8, 10, 11].

Another comment is that the reflective property (3.1) depends on the activation function $\sigma$, while conditions in previous works [1, 2, 8, 10, 11] are all defined for the target function $f^*$ or $h^*$ itself. There does exist a variant of our reflective property that is independent of $\sigma'$, namely,

$$\mathbb{E}_{z_S \sim \mathcal{N}(0, I_S)} [h^*(z) z_S] = 0, \quad \forall \, z_S^\perp, \tag{3.2}$$

which actually implies (3.1). But these two conditions are different: $h^*(z) = z_1 + z_1 z_2 + z_1 z_2 z_3$ does not satisfy (3.2) but still satisfies (3.1) if $\sigma(\zeta) = \zeta$. We use (3.1) with $\sigma'$ because we want to emphasize that the SGD learnability depends on the activation function $\sigma$. If $\sigma$ is less expressive, then SGD may not learn the target function even if $h^*$ itself satisfies the merged-staircase property. Typically people use activation functions that are expressive enough, for which (3.1) and (3.2) are similar. In addition, it can be verified that (3.2) with some nontrivial $S$ is equivalent to $\text{isoLeap}(h^*) \geq 2$ that means $h^* : V \to \mathbb{R}$ does not satisfy the merged-staircase property for some orthonormal basis of $V$ [2], and the idea of leaps is used in [8, 10]. We include the proof of equivalence in Appendix B.1.

Our main result in this section is that the reflective property with nontrivial $S$ would lead to a positive lower bound of $\mathcal{E}(\rho_t)$ along the training dynamics, which provides a necessary condition for the SGD-learnability and is formally stated as follows.

**Theorem 3.4.** *Suppose that Assumption 3.1 holds with $\rho_w \sim \mathcal{N}(0, \frac{1}{d} I_d)$, and that $h^* : V \to \mathbb{R}$ satisfies the reflective property with respect to some subspace $S \subset V$ and activation function $\sigma$. Then for any $T > 0$, there exists a constant $C > 0$ depending only on $p$, $h^*$, $K_\sigma$, $K_\xi$, $K_\rho$, and $T$, such that*

$$\inf_{0 \leq t \leq T} \mathcal{E}(\rho_t) \geq \frac{1}{2} \mathbb{E}_{z \sim \mathcal{N}(0, I_V)} \left[ |h^*(z) - h_{S^\perp}^*(z_S^\perp)|^2 \right] - \frac{C}{d^{1/2}}, \tag{3.3}$$

*where $h_{S^\perp}^*(z_S^\perp) = \mathbb{E}_{z_S}[h^*(z)]$. In particular, if $h^*(z)$ is not independent of $z_S$, then for any $T > 0$, there exists $d(T) > 0$ depending only on $p$, $h^*$, $K_\sigma$, $K_\xi$, $K_\rho$, and $T$, such that for any $d > d(T)$, we have*

$$\inf_{0 \leq t \leq T} \mathcal{E}(\rho_t) \geq \frac{1}{4} \mathbb{E}_{z \sim \mathcal{N}(0, I_V)} \left[ |h^*(z) - h_{S^\perp}^*(z_S^\perp)|^2 \right] > 0. \tag{3.4}$$

It is worth remarking that in Theorem 3.4, the training time $T$ is a constant independent of the dimension $d$. If a longer $d$-dependent training beyond a constant time is allowed, then $\mathcal{E}(\rho_t)$ might be reasonably small even if the necessary condition is not satisfied, see e.g. [2, 18, 26].

In Appendix B.2, we include a brief discussion of the sample complexity result of SGD implied by Theorem 3.4. In particular, SGD with $\mathcal{O}(d)$ samples cannot recover $f^*$ reliably if the refelctive

property holds, which is consistent with observations in previous works such as [1, 2]. We also remark that our result in Theorem 3.4 is established for the mean-field dynamics corresponding to the one-pass SGD (2.3), any may not apply for other variants of SGD. In particular, some recent works [6, 11, 16] prove that multi-pass SGD with batch-reuse mechanism can learn some target functions with fewer samples than one-pass SGD.

## 3.2   Proof Sketch for Theorem 3.4

To prove Theorem 3.4, the main intuition is that under some mild assumptions, if (3.1) is satisfied and the initial distribution $\rho_0$ is supported in $\{(a, w) \in \mathbb{R}^{d+1} : w_S = 0\}$, where $w_S$ is the orthogonal projection of $w \in \mathbb{R}^d$ onto $S$, then $\rho_t$ is supported in $\{(a, w) \in \mathbb{R}^{d+1} : w_S = 0\}$ for all $t \geq 0$. This means that the trained neural network $f_{\mathrm{NN}}(x; \rho_t)$ learns no information about $x_S$, the orthogonal projection of $x \in \mathbb{R}^d$ onto $S$, and hence cannot approximate $f^*(x) = h^*(x_V)$ with arbitrarily small error if $h^*(z)$ is dependent on $z_S$. We formulate this observation in the following theorem.

**Theorem 3.5.** *Suppose that Assumption 3.1 hold and let $\rho_t$ be the solution to (2.4). Let $S \subset \mathbb{R}^d$ be a subspace with the projection map $\mathcal{P}_S : \mathbb{R}^{d+1} \to S$ that maps $(a, w)$ to $w_S$. If $(\mathcal{P}_S)_{\#}\rho_0 = \delta_S$, where $\delta_S$ is the delta measure on $S$ and*

$$\mathbb{E}_x \left[ f^*(x)\sigma' \left( w^\top x_S^\perp \right) x_S \right] = 0, \quad \forall\, w \in \mathbb{R}^d, \tag{3.5}$$

*where $x_S^\perp = x - x_S$, then it holds for any $t \geq 0$ that*

$$(\mathcal{P}_S)_{\#}\rho_t = \delta_S. \tag{3.6}$$

Here, the delta measure $\delta_S$ on $S$ is a probability measure on $S$ such that for any continuous and compactly supported function $\varphi : S \to \mathbb{R}$, it holds that $\int_S \varphi(x)\delta_S(dx) = \varphi(0)$. In Theorem 3.5, the condition (3.5) is stated in terms of $f^*$. We will show later that it is closely related to and is actually implied by (3.1), via a decomposition $w^\top x_S^\perp = w^\top (x - x_V) + w^\top (x_V - x_S)$, with $w^\top (x - x_V)$ and $w^\top (x_V - x_S)$ corresponding to $u$ and $v^\top z_S^\top$ in (3.1), respectively. The main idea in the proof of Theorem 3.5 is to construct a flow $\hat{\rho}_t$ in the space $\mathcal{P}(\mathbb{R} \times S^\perp)$, where $S^\perp$ is the orthogonal complement of $S$ in $\mathbb{R}^d$, and then show that $\rho_t = \hat{\rho}_t \times \delta_S$ is the solution to (2.4). More specifically, the flow $\hat{\rho}_t$ is constructed as the solution to the following evolution equation in $\mathcal{P}(\mathbb{R} \times S^\perp)$:

$$\begin{cases} \partial_t \hat{\rho}_t = \nabla_{\hat{\theta}} \cdot \left( \hat{\rho}_t \hat{\xi}(t) \nabla_{\hat{\theta}} \hat{\Phi}(\hat{\theta}; \hat{\rho}_t) \right), \\ \hat{\rho}_t\big|_{t=0} = \hat{\rho}_0, \end{cases} \tag{3.7}$$

where $\hat{\rho}_0 \in \mathcal{P}(\mathbb{R} \times S^\perp)$ satisfies $\rho_0 = \hat{\rho}_0 \times \delta_S$, $\hat{\theta} = (a, w_S^\perp)$, $\hat{\xi}(t) = \mathrm{diag}(\xi_a(t), \xi_w(t)I_{S^\perp})$, and

$$\hat{\Phi}(\hat{\theta}; \hat{\rho}) = a\mathbb{E}_x \left[ \left( \hat{f}_{\mathrm{NN}}(x_S^\perp; \hat{\rho}) - f^*(x) \right) \sigma \left( (w_S^\perp)^\top x_S^\perp \right) \right] = \hat{V}(\hat{\theta}) + \int \hat{U}(\hat{\theta}, \hat{\theta}')\hat{\rho}(d\hat{\theta}'),$$

with

$$\hat{f}_{\mathrm{NN}}(x_S^\perp; \hat{\rho}) = \int a\sigma \left( (w_S^\perp)^\top x_S^\perp \right) \hat{\rho}(da, dw_S^\perp),$$

and

$$\hat{V}(\hat{\theta}) = -a\mathbb{E}_x \left[ f^*(x)\sigma \left( (w_S^\perp)^\top x_S^\perp \right) \right], \quad \hat{U}(\theta, \theta') = aa'\mathbb{E}_x \left[ \sigma \left( (w_S^\perp)^\top x_S^\perp \right) \sigma \left( ((w_S^\perp)')^\top x_S^\perp \right) \right].$$

The detailed proof will be presented in Appendix A.1.

In practice, both $V$ and $S$ are unknown and it is nontrivial to choose an initialization $\rho_0$ supported in $\{(a, w) \in \mathbb{R}^{d+1} : w_S = 0\}$. However, one can set $\rho_0 = \rho_a \times \rho_w$ with $\rho_w \sim \mathcal{N}(0, \frac{1}{d}I_d)$ and this can make the marginal distribution of $\rho_0$ on $S$ very close to the the delta measure $\delta_S$ if $d >> p = \dim(V) \geq \dim(S)$, which fits the setting of subspace-sparse polynomials. Rigorously, we have the following theorem stating dimension-free stability with respect to initial distribution, with the proof deferred to Appendix A.2.

**Theorem 3.6.** *Suppose that Assumption 3.1 holds for both $\rho_0$ and $\tilde{\rho}_0$. Let $\rho_t$ solve $\partial_t \rho_t = \nabla_\theta \cdot (\rho_t \xi(t)\nabla_\theta \Phi(\theta; \rho_t))$ and let $\tilde{\rho}_t$ solve $\partial_t \tilde{\rho}_t = \nabla_\theta \cdot (\tilde{\rho}_t \xi(t)\nabla_\theta \Phi(\theta; \tilde{\rho}_t))$. Then for any $T \in (0, +\infty)$, there exists a constant $C_s > 0$ depending only on $p$, $h^*$, $K_\sigma$, $K_\xi$, $K_\rho$, and $T$, such that*

$$\sup_{0 \leq t \leq T} \mathbb{E}_x \left[ |f_{NN}(x; \rho_t) - f_{NN}(x; \tilde{\rho}_t)|^2 \right] \leq C_s W_2^2(\rho_0, \tilde{\rho}_0), \tag{3.8}$$

*where $W_2(\cdot, \cdot)$ is the 2-Wasserstein metric.*

Based on Theorem 3.5 and Theorem 3.6, Theorem 3.4 can be proved by some straightforward computation, for which the details can be found in Appendix A.3.

**Algorithm 1** Training strategy
___
1: Set the initial distribution as $\rho_0 = \rho_a \times \delta_{\mathbb{R}^d}$, where $\rho_a = \mathcal{U}([-1,1])$ and $\delta_{\mathbb{R}^d}$ is the delta measure on $\mathbb{R}^d$.
2: Set $\xi_a(t) = 0$ and $\xi_w(t) = 1$ for $0 \le t \le T$, and train the neural network with activation function $\sigma$. Denote by $(a, w(a, t))$, $0 \le t \le T$ the trajectory of a single particle that starts at $(a, 0)$.
3: Repeat Step 2 for $p$ times independently and obtain $p$ copies of parameters at $T$, say $(a_i, w(a_i, T))$ with $a_i \sim \mathcal{U}([-1,1])$, $i = 1, 2, \ldots, p$.
4: Reset $\rho_T$ as the distribution of $(0, u(a_1, \ldots, a_p, T)) = \left(0, \frac{1}{p}\sum_{i=1}^{p} w(a_i, T)\right)$. Train the neural network with $\xi_a(t) = 1$, $\xi_w(t) = 0$, and a new activation function $\hat{\sigma}(\zeta) = (1 + \zeta)^n$, where $n = \deg(f^*)$, for $t \ge T$ starting at $\rho_T$.
___

## 4 Sufficient Condition for SGD-Learnability

In this section, we propose a sufficient condition and a training strategy that can guarantee the exponential decay of $\mathcal{E}(\rho_t)$ with constants independent of the dimension $d$.

### 4.1 Training Procedure and Convergence Guarantee

We prove in Section 3 that if the trained parameters always stay in a proper subspace $\{(a, w) \in \mathbb{R}^{d+1} : w_S = 0\}$, then $f_{\mathrm{NN}}(x; \rho_t)$ cannot learn all information about $f^*$ or $h^*$. Ideally, one would expect the negation to be a sufficient condition for the SGD-learnability, i.e., the existence of a choice of learning rates and initial distribution that guarantees $\lim_{t\to\infty} \mathcal{E}(\rho_t)$ with dimension-free rate. This is almost true but we need a slightly stronger condition due to technical issues. More specifically, we need that the Taylor's expansion of some dynamics (not the dynamics itself) is not trapped in any proper subspace.

**Assumption 4.1.** *Consider the following flow $\hat{w}_V(t)$ in $V$:*

$$\begin{cases} \frac{d}{dt}\hat{w}_V(t) = \mathbb{E}_z\left[z h^*(z)\sigma'(\hat{w}_V(t)^\top z)\right], \\ \hat{w}_V(0) = 0. \end{cases} \tag{4.1}$$

*We assume that for some $s \in \mathbb{N}_+$, the Taylor's expansion up to $s$-th order of $\hat{w}_V(t)$ at $t = 0$ is not contained in any proper subspace of $V$.*

Assumption 4.1 aims to state the same observation as the merged-staircase property in [1]. As a simple example, if $V = \mathbb{R}^p$ and $h^*(z) = z_1 + z_1 z_2 + z_1 z_2 z_3 + \cdots + z_1 z_2 \cdots z_p$ which satisfies the merged-staircase property, then it can be computed that the leading order terms of the coordinates of $\hat{w}_V(t)$ are given by $(c_1 t, c_2 t^2, c_3 t^{2^2}, \ldots, c_p t^{2^{p-1}})$ with nonzero constants $c_1, c_2, \ldots, c_p$ if $\sigma \in \mathcal{C}^s(\mathbb{R})$ with $s = 2^{p-1}$ and $\sigma^{(1)}(0), \sigma^{(2)}(0), \ldots, \sigma^{(p)}(0)$ are all nonzero (see Proposition 33 in [1]). This is to say that Assumption 4.1 with $s = 2^{p-1}$ is satisfied for this example. We provide further characterization of Assumption 4.1 by verifying it in a more general setting in Appendix D.1.

We also remark that the Taylor's expansion of the flow $\hat{w}_V(t)$ that solves (4.1) depends only on the $h^*$ and $\sigma^{(1)}(0), \sigma^{(2)}(0), \ldots, \sigma^{(s)}(0)$. We require some additional regularity assumption on higher-order derivatives of $\sigma$.

**Assumption 4.2.** *Assume that $\sigma$ satisfies $\sigma \in \mathcal{C}^{L+1}(\mathbb{R})$ and $\sigma, \sigma', \sigma'', \sigma^{(L+1)} \in L^\infty(\mathbb{R})$, where $L = 2sn\binom{n+p}{p}$ with $n = deg(f^*) = deg(h^*)$ and $s$ being the positive integer in Assumption 4.1.*

Our proposed training strategy is stated in Algorithm 1. The training strategy is inspired by the two-stage strategy proposed in [1] that trains the parameters $w$ with fixed $a$ for $t \in [0, T]$ and then trains the parameter $a$ with fixed $w$ and a perturbed activation function for $t \ge T$. Several important modifications are made since we consider general polynomials, rather than polynomials on hypercubes as in [1]. In particular,

- We need to repeat Step 2 (training $w$) for $p$ times and use their average as the initialization of training $a$, while this step only needs to be done once in [1]. The reason is that the space of polynomials on the hypercube $\{\pm 1\}^p$ is essentially a linear space with dimension $2^p$. However, the space of general polynomials on $V$ is an $\mathbb{R}$-algebra that is also a linear

space but is of infinite dimension. Therefore, to make the kernel matrix in training $a$ non-degenerate, we require some algebraic independence which can be guaranteed by $u(a_1, \ldots, a_p), t)) = \frac{1}{p} \sum_{i=1}^{p} w(a_i, t)$, $0 < t \leq T$, though linear independence suffices for [1]. Let us also emphasize that each run of Step 2 involves training an interacting particle system instead of training a single particle.

- In Step 4, we use a new activation function $\hat{\sigma}(\zeta) = (1 + \zeta)^n$ that is a polynomial of the same degree as $f^*$ and $h^*$. The reason is still that we work with the space general polynomials whose dimension as a linear space is infinite. Thus, we need the specific form $\hat{\sigma}(\zeta) = (1 + \zeta)^n$ to guarantee the trained neural network $f_{\text{NN}}(x; \rho_t)$ is a polynomial with degree at most $n = \deg(f^*) = \deg(h^*)$. As a comparison, in the setting of [1], all functions on $\{\pm 1\}^p$ can be understood as a polynomial, and no specific format of the new activation function is needed.

Our main theorem in this section is as follows, stating that the loss functional $\mathcal{E}(\rho_t)$ can decay to 0 exponentially fast, with rates independent of the dimension $d$.

**Theorem 4.3.** *Suppose that Assumption 4.1 and 4.2 hold and let $\rho_t$ be the flow generated by Algorithm 1. There exist constants $C_1, C_2 > 0$ depending on $h^*, \sigma, n, p, s$, such that*

$$\mathcal{E}(\rho_t) \leq C_1 \exp(-C_2 t), \quad \forall\, t \geq 0.$$

Let us also remark that it is possible to use the original dynamics $\hat{w}_V$ defined in (4.1) when we state Assumption 4.1, which can actually imply its Taylor's expansion up to some order is not trapped in any proper subspace of $V$ if we further assume $\hat{w}_V(t)$ is analytic. We choose to directly use Taylor's expansion in Assumption 4.1 since we want to avoid the additional analytic assumption and to emphasize that the constants $C_1, C_2$ in Theorem 4.3 depend on the order $s$ of the Tayler's expansion satisfying Assumption 4.1.

Discussion about the sample complexity implied by Theorem 4.3 is included in Appendix D.2, suggesting that $\mathcal{O}(d)$ samples suffices for SGD to learn $f^*$ reliably if conditions in Theorem 4.3 are true. This is also consistent with previous works such as [1, 2].

## 4.2 Proof Sketch for Theorem 4.3

To prove Theorem 4.3 we follow the same general strategy as [1], though some technical analysis is significantly different due to the roatation-invariant setting. The main goal here is to show before Step 4, the algorithm already learned a diverse set of features. After that, note that Step 4 in Algorithm 1 is essentially a convex/quadratic optimization problem (since we only train $a$ and set $\xi_w(t) = 0$). In addition, thanks to the new activation function $\hat{\sigma}(\zeta) = (1 + \zeta)^n$, one only needs to consider $\mathbb{P}_{V,n}$ that is the space of of all polynomials on $V$ with degree at most $n = \deg(h^*) = \deg(f^*)$. The dimension of $\mathbb{P}_{V,n}$ as a linear space is $\binom{n+p}{p}$. Let $p_1, p_2, \ldots, p_{\binom{n+p}{p}}$ be the orthonormal basis of $\mathbb{P}_{V,n}$ with input $z \sim \mathcal{N}(0, I_V)$ and define the kernel matrix

$$\mathcal{K}_{i_1, i_2}(t) = \mathbb{E}_{a_1, \ldots, a_p} \left[ \mathbb{E}_{z, z'} \left[ p_{i_1}(z) \hat{\sigma}(u(a_1, \ldots, a_p, t)^\top z) \hat{\sigma}(u(a_1, \ldots, a_p, t)^\top z') p_{i_2}(z') \right] \right], \quad (4.2)$$

where $\hat{\sigma}(\xi) = (1 + \xi)^n$, $(a_1, \ldots, a_p) \sim \mathcal{U}([-1, 1]^p)$, $1 \leq i_1, i_2 \leq \binom{n+p}{p}$, and $0 \leq t \leq T$. As long as this kernel matrix is non-degenerate, we know that the loss functional is strongly convex with respect to the parameters in the second layer when fixing the first layer, and thus, it can be computed straightforwardly that the loss decays to 0 exponentially fast for $t \geq T$, leading to the desired convergence rate in Theorem 4.3; see Appendix C.3 for details. Thus, the main part in the proof of Theorem 4.3 is to establish the non-degeneracy of the kernel matrix.

**Proposition 4.4.** *Suppose that Assumption 4.1 and 4.2 hold. There exist constants $C, T > 0$ depending on $h^*, \sigma, n, p, s$, such that*

$$\lambda_{\min}(\mathcal{K}(t)) \geq C t^{2sn \binom{n+p}{p}}, \quad \forall\, 0 \leq t \leq T, \quad (4.3)$$

*where $\lambda_{\min}(\mathcal{K}(t))$ is the smallest eigenvalue of $\mathcal{K}(t)$.*

In the rest of this subsection, we sketch the main ideas in the proof of Proposition 4.4, with the details of the proof being deferred to Appendix C. We first show that $w(a_i, t)$ and $u(a_1, \ldots, a_p, t)$ can be

approximated well by $\hat{w}(a_i, t)$ and $\hat{u}(a_1, \ldots, a_p, t) = \frac{1}{p} \sum_{i=1}^{p} \hat{w}(a_i, t)$ that are polynomials in $a_i$ and $a_1, \ldots, a_p$ respectively. This approximation step follows [1] closely and is analyzed detailedly in Appendix C.1. Therefore, to give a positive lower bound of $\lambda_{\min}(\mathcal{K}(t))$, one only needs to show the non-degeneracy of the matrix $\hat{M}(\mathbf{a}, t) \in \mathbb{R}^{\binom{n+p}{p} \times \binom{n+p}{p}}$ with

$$\hat{M}_{i_1, i_2}(\mathbf{a}, t) = \mathbb{E}_z \left[ p_{i_1}(z) \hat{\sigma}(\hat{u}(\mathbf{a}_{i_2}, t)^\top z) \right],$$

where $\mathbf{a} = \left( \mathbf{a}_1, \mathbf{a}_2, \ldots, \mathbf{a}_{\binom{n+p}{p}} \right)$ and $\mathbf{a}_i \in \mathbb{R}^p$ for $i = 1, 2, \ldots, \binom{n+p}{p}$. Intuitively, this non-degeneracy can be implied by

$$\text{span} \left\{ \hat{\sigma}(\hat{u}(a_1, \ldots, a_p, t)^\top z) : a_1, a_2, \ldots, a_p \in [-1, 1] \right\} = \mathbb{P}_{V,n},$$

which is true if $\hat{u}_i(a_1, \ldots, a_p, t)$, $1 \le i \le p$ are $\mathbb{R}$-algebraically independent polynomials in $a_1, a_2, \ldots, a_p$, where $\hat{u}_i$ is the $i$-th coefficient of $\hat{u}$ under some basis of $V$, and algebraic independence can be obtained from linear independence by taking the average of independent copies. We illustrate this intuition with a bit more detail.

**Algebraic independence of $\hat{u}_i$.** With Assumption 4.1, $\hat{w}_1(a, t), \hat{w}_2(a, t), \ldots, \hat{w}_1(a, t)$ can be proved as $\mathbb{R}$-linear independent polynomials in $a \in \mathbb{R}$. Then one can apply the following theorem to boost the linear independence of $\hat{w}_i$, whose constant term is zero since initialization in training is set as $\rho_0 = \rho_a \times \delta_{\mathbb{R}^d}$, to the algebraic independence of $\hat{u}_i$.

**Theorem 4.5.** *Let $v_1, v_2, \ldots, v_p \in \mathbb{R}[a]$ be $\mathbb{R}$-linearly independent polynomials with the constant terms being zero. Then $\frac{1}{p}(v_1(a_1) + \cdots + v_1(a_p)), \ldots, \frac{1}{p}(v_p(a_1) + \cdots + v_p(a_p)) \in \mathbb{R}[a_1, a_2, \ldots, a_p]$ are $\mathbb{R}$-algebraically independent.*

The proof of Theorem 4.5 is deferred to Appendix C.2 and is based on the celebrated Jacobian criterion stated as follows.

**Theorem 4.6** (Jacobian criterion [7])**.** *$v_1, v_2, \ldots, v_p \in \mathbb{R}[a_1, a_2, \ldots, a_p]$ are $\mathbb{R}$-algebraically independent if and only if*

$$\det \begin{pmatrix} \frac{\partial v_1}{\partial a_1} & \frac{\partial v_1}{\partial a_2} & \cdots & \frac{\partial v_1}{\partial a_p} \\ \frac{\partial v_2}{\partial a_1} & \frac{\partial v_2}{\partial a_2} & \cdots & \frac{\partial v_2}{\partial a_p} \\ \vdots & \vdots & \ddots & \vdots \\ \frac{\partial v_p}{\partial a_1} & \frac{\partial v_p}{\partial a_2} & \cdots & \frac{\partial v_p}{\partial a_p} \end{pmatrix}$$

*is a nonzero polynomial in $\mathbb{R}[a_1, a_2, \ldots, a_p]$.*

**Non-degeneracy of $\hat{M}(\mathbf{a}, t)$.** With the observation that $\text{span} \left\{ \hat{\sigma}(q^\top z) : q \in V \right\} = \mathbb{P}_{V,n}$ (see Lemma C.13), we define another matrix $X(\mathbf{q}) \in \mathbb{R}^{\binom{n+p}{p} \times \binom{n+p}{p}}$ via

$$X_{i_1, i_2}(\mathbf{q}) = \mathbb{E}_z \left[ p_{i_1}(z) \sigma(\mathbf{q}_{i_2}^\top z) \right],$$

where $\mathbf{q} = \left( \mathbf{q}_1, \mathbf{q}_2, \ldots, \mathbf{q}_{\binom{n+p}{p}} \right)$ with $\mathbf{q}_i \in V$, and prove that $\det(X(\mathbf{q}))$ is a non-zero polynomial in $\mathbf{q}$ of the form

$$\det(X(\mathbf{q})) = \sum_{i=1}^{\binom{n+p}{p}} \sum_{0 \le \|\mathbf{j}_i\|_1 \le n} X_{\mathbf{j}} \mathbf{q}^{\mathbf{j}} = \sum_{i=1}^{\binom{n+p}{p}} \sum_{0 \le \|\mathbf{j}_i\|_1 \le n} X_{\mathbf{j}} \prod_{l=1}^{\binom{n+p}{p}} \mathbf{q}_l^{\mathbf{j}_l},$$

where $\mathbf{q}_i$ is understood as a (coefficient) vector in $\mathbb{R}^p$ associated with a fixed orthonormal basis of $V$ and $\mathbf{j} = \left( \mathbf{j}_1, \mathbf{j}_2, \ldots, \mathbf{j}_{\binom{n+p}{p}} \right)$ with $\mathbf{j}_i \in \mathbb{N}^p$. Then setting $\mathbf{q} = \hat{u}(\mathbf{a}_{i_2}, t)$ leads to

$$\det(\hat{M}(\mathbf{a}, t)) = \sum_{i=1}^{\binom{n+p}{p}} \sum_{0 \le \|\mathbf{j}_i\|_1 \le n} X_{\mathbf{j}} \prod_{l=1}^{\binom{n+p}{p}} \hat{u}(\mathbf{a}_l, t)^{\mathbf{j}_l}.$$

To prove that $\det(\hat{M}(\mathbf{a}, t))$ is a non-zero polynomial in $\mathbf{a}$, i.e., $\hat{M}(\mathbf{a}, t)$ is non-degenerate, we use the following lemma linking algebraic independence back to linear independence.

**Lemma 4.7.** *Suppose that* $v_1, v_2, \ldots, v_p \in \mathbb{R}[a_1, a_2, \ldots, a_p]$ *are* $\mathbb{R}$-*algebraically independent. For any* $m \geq 1$*, the following polynomials in* $\mathbf{a} = (\mathbf{a}_1, \mathbf{a}_2, \ldots, \mathbf{a}_m) \in (\mathbb{R}^p)^m$ *are* $\mathbb{R}$-*linearly independent*

$$\prod_{l=1}^{m} \mathbf{v}(\mathbf{a}_l)^{\mathbf{j}_l}, \quad 1 \leq \|\mathbf{j}_i\|_1 \leq n, \ 1 \leq i \leq m,$$

*where* $\mathbf{v} = (v_1, v_2, \ldots, v_p)$.

The proof of Lemma 4.7 and some other related analysis are deferred to Appendix C.3.

## 5 Conclusion and Discussions

In this work, we generalize the merged-staircase property in [1] to a basis-free version and establish a necessary condition for learning a subspace-sparse polynomial on Gaussian input with arbitrarily small error. Moreover, we prove the exponential decay property of the loss functional under a sufficient condition that is slightly stronger than the necessary one. The bounds and rates are all dimension-free. Our work provides some understanding of the mean-field dynamics, though its general behavior is extremely difficult to characterize due to the non-convexity of the loss functional.

Let us also make some comments on limitations and future directions. Firstly, there is still a gap between the necessary condition and the sufficient condition, which is basically from the fact that the sufficient condition is built on the Taylor's expansion of the flow (4.1). One future research question is whether we can fill the gap by considering the original flow (4.1) rather than its Taylor's expansion. Secondly, Algorithm 1 repeats training $w$ for $p$ times and takes the average of parameters, which is different from the usual strategy for training neural networks. This step is used to guarantee the algebraic independence. We conjecture that this step can be removed since the general algebraic independence is too strong when we have some preknowledge on the degree of $f^*$ or $h^*$, which deserves future research.

## Acknowledgments and Disclosure of Funding

The work of R. Ge is supported by NSF Award DMS-2031849 and CCF-1845171 (CAREER). We thank Joan Bruna for helpful comments and discussion.

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

# A Proofs for Section 3

This section collects the proofs of Theorem 3.5, Theorem 3.6, and Theorem 3.4.

## A.1 Proof of Theorem 3.5

**Existence and uniqueness of solutions to** (2.4)    Before proving Theorem 3.5, let us remark on the existence and uniqueness of solution to the mean-field dynamics (2.4). According to Remark 7.1 in [20] and Theorem 1.1 in [27], the PDE (2.4) admits a unique solution if Assumption 3.1 (ii) holds and both $\nabla V(\theta)$ and $\nabla_\theta U(\theta, \theta')$ are bounded and Lipschitz continous. Here we recall that $V$ and $U$ are defined in (2.5). With Assumption 3.1 (i) and (iii), it is not hard to verify the boundedness and Lipschitz continuity of $\nabla V(\theta)$ and $\nabla_\theta U(\theta, \theta')$ by noticing that any finite-order moment of $\mathcal{N}(0, I_d)$ is finite.

*Proof of Theorem 3.5.*  Since $(\mathcal{P}_S)_\# \rho_0 = \delta_S$, the initial distribution $\rho_0$ can be decomposed as $\rho_0 = \hat\rho_0 \times \delta_S$, where $\hat\rho_0 \in \mathcal{P}(\mathbb{R} \times S^\perp)$. Consider the following evolution equation (3.7) in $\mathcal{P}(\mathbb{R} \times S^\perp)$. By the discussion at the beginning of Section A.1, we know that $\rho_t$ is the unique solution to (2.4). Similar arguments also leads to the existence and uniquess of the solution to (3.7).

We will show that the solution to (2.4) must be of the form

$$\rho_t = \hat\rho_t \times \delta_S, \tag{A.1}$$

where $\hat\rho_t$ solves (3.7), and this decomposition can immediatel imply (3.6). By the uniquess of the solution, it suffices to verify that $\rho_t = \hat\rho_t \times \delta_S$ is a solution to (2.4). It follows directly from (A.1) that

$$f_{\text{NN}}(x; \rho_t) = f_{\text{NN}}(x; \hat\rho_t \times \delta_S) = \hat{f}_{\text{NN}}(x_S^\perp; \hat\rho_t),$$

and hence that

$$\partial_a \Phi(\theta; \rho_t) = \partial_a \hat\Phi(\hat\theta; \hat\rho_t), \quad \text{if } w_S = 0. \tag{A.2}$$

We also have that

$$\begin{aligned}
\nabla_{w_S^\perp} \Phi(\theta; \rho_t) &= a \mathbb{E}_x \left[ (f_{\text{NN}}(x; \rho_t) - f^*(x)) \sigma'(w^\top x) x_S^\perp \right] \\
&= a \mathbb{E}_x \left[ \left( \hat{f}_{\text{NN}}(x_S^\perp; \hat\rho_t) - f^*(x) \right) \sigma' \left( (w_S^\perp)^\top x_S^\perp \right) x_S^\perp \right] \\
&= \nabla_{w_S^\perp} \hat\Phi(\hat\theta; \hat\rho_t),
\end{aligned} \tag{A.3}$$

if $w_S = 0$. In addition, it holds also for $w_S = 0$ that

$$\begin{aligned}
\nabla_{w_S} \Phi(\theta; \rho_t) &= a \mathbb{E}_x \left[ (f_{\text{NN}}(x; \rho_t) - f^*(x)) \sigma'(w^\top x) x_S \right] \\
&= a \mathbb{E}_x \left[ \hat{f}_{\text{NN}}(x_S^\perp; \hat\rho_t) \sigma' \left( (w_S^\perp)^\top x_S^\perp \right) x_S \right] - a \mathbb{E}_x \left[ f^*(x) \sigma' \left( (w_S^\perp)^\top x_S^\perp \right) x_S \right] \tag{A.4} \\
&= 0,
\end{aligned}$$

where we used $\mathbb{E}_{x_S}[x_S] = 0$ and (3.5). Combining (A.2), (A.3), and (A.4), we have for any $\eta \in \mathcal{C}_c^\infty(\mathbb{R}^{d+1} \times (0, +\infty)) = \mathcal{C}_c^\infty(\mathbb{R} \times S^\perp \times S \times (0, +\infty))$ that

$$\begin{aligned}
&\iint \left( -\partial_t \eta + \nabla_\theta \eta \cdot (\xi(t) \nabla_\theta \Phi(\theta; \rho_t)) \right) \rho_t(d\theta) dt \\
&= \iiint \Big( -\partial_t \eta(\theta, t) + \xi_a(t) \partial_a \eta(\theta, t) \cdot \partial_a \Phi(\theta; \rho_t) + \xi_w(t) \nabla_{w_S^\perp} \eta(\theta, t) \cdot \nabla_{w_S^\perp} \Phi(\theta; \rho_t) \\
&\qquad\qquad + \xi_w(t) \nabla_{w_S} \eta(\theta, t) \cdot \nabla_{w_S} \Phi(\theta; \rho_t) \Big) \hat\rho_t(d\hat\theta) \delta_S(dw_S) dt \\
&= \iint \Big( -\partial_t \eta(\hat\theta, 0, t) + \xi_a(t) \partial_a \eta(\hat\theta, 0, t) \cdot \partial_a \hat\Phi(\hat\theta; \hat\rho_t) \\
&\qquad\qquad + \xi_w(t) \nabla_{w_S^\perp} \eta(\hat\theta, 0, t) \cdot \nabla_{w_S^\perp} \hat\Phi(\hat\theta; \hat\rho_t) \Big) \hat\rho_t(d\hat\theta) dt \\
&= \iint \left( -\partial_t \eta(\hat\theta, 0, t) + \nabla_{\hat\theta} \eta(\hat\theta, 0, t) \cdot (\hat\xi(t) \nabla_{\hat\theta} \hat\Phi(\hat\theta; \hat\rho_t)) \right) \hat\rho_t(d\hat\theta) dt \\
&= 0,
\end{aligned}$$

where the last equality holds by applying the test function $\eta(\cdot, 0, \cdot) \in \mathcal{C}_c^\infty(\mathbb{R} \times S^\perp \times (0, +\infty))$ to (3.7). The proof is hence completed. □

## A.2 Proof of Theorem 3.6

The proof of Theorem 3.6 uses some ideas from the proof of Theorem 16 in [1]. Similar ideas also exist in earlier works (see e.g., [19, 20]).

**Lemma A.1.** *Suppose that Assumption 3.1 holds and let $\rho_t$ solve (2.4). Then for any $t > 0$, $\rho_t$ is supported in $\{\theta = (a, w) \in \mathbb{R}^{d+1} : |a| \leq K_\rho + K_\xi K_\sigma \mathbb{E}_z \left[|h^*(z)|^2\right]^{1/2} t\}$.*

*Proof.* The particle dynamics for $a_t$ associated with (2.4) is given by

$$\frac{d}{dt} a_t = \xi_a(t) \mathbb{E}_x \left[(f_{\mathrm{NN}}(x; \rho_t) - f^*(x))\sigma(w_t^\top x)\right],$$

which implies that

$$\left|\frac{d}{dt} a_t\right| \leq K_\xi K_\sigma \left|\mathbb{E}_x \left[f_{\mathrm{NN}}(x; \rho_t) - f^*(x)\right]\right| \leq K_\xi K_\sigma (2\mathcal{E}(\rho_0))^{1/2} = K_\xi K_\sigma \mathbb{E}_z \left[|h^*(z)|^2\right]^{1/2}.$$

Therefore, one has $|a_t| \leq |a_0| + K_\xi K_\sigma \mathbb{E}_z \left[|h^*(z)|^2\right]^{1/2} t$, which completes the proof. $\square$

**Lemma A.2.** *Suppose that Assumption 3.1 holds for both $\rho_0$ and $\tilde{\rho}_0$. Let $\rho_t$ solve $\partial_t \rho_t = \nabla_\theta \cdot (\rho_t \xi(t) \nabla_\theta \Phi(\theta; \rho_t))$ and let $\tilde{\rho}_t$ solve $\partial_t \tilde{\rho}_t = \nabla_\theta \cdot (\tilde{\rho}_t \xi(t) \nabla_\theta \Phi(\theta; \tilde{\rho}_t))$. For any coupling $\gamma_0 \in \Gamma(\rho_0, \tilde{\rho}_0)$, let $\gamma_t \in \Gamma(\rho_t, \tilde{\rho}_t)$ be the associated coupling during the evolution and define*

$$\Delta(t) = \iint \left(|a - \tilde{a}|^2 + \|w - \tilde{w}\|^2\right) \gamma_t(d\theta, d\tilde{\theta}).$$

*Then it holds for any $0 \leq t \leq T$ that*

$$\mathbb{E}_x \left[|f_{NN}(x; \rho_t) - f_{NN}(x; \tilde{\rho}_t)|^2\right] \leq C_f \Delta(t), \tag{A.5}$$

*and*

$$\frac{d}{dt} \Delta(t) \leq C_\Delta \Delta(t), \tag{A.6}$$

*where $C_f$ and $C_\Delta$ are constants depending only on $p$, $h^*$, $K_\sigma$, $K_\xi$, $K_\rho$, and $T$.*

*Proof of Theorem 3.6.* Let $C_f$ and $C_\Delta$ be the constants in Lemma A.2. For any $\epsilon > 0$, there exists a coupling $\gamma_0 \in \Gamma(\rho_0, \tilde{\rho}_0)$ such that

$$\iint (|a - \tilde{a}|^2 + \|w - \tilde{w}\|^2)\gamma_0(d\theta, d\tilde{\theta}) \leq W_2^2(\rho_0, \tilde{\rho}_0) + \epsilon.$$

Define $\gamma_t \in \Gamma(\rho_t, \tilde{\rho}_t)$ and $\Delta(t)$ as in Lemma A.2. According (A.6) and the Grönwall's inequality, it holds that

$$\Delta(t) \leq \Delta(0) e^{C_\Delta t} \leq \left(W_2^2(\rho_0, \tilde{\rho}_0) + \epsilon\right) e^{C_\Delta t}, \quad \forall 0 \leq t \leq T,$$

which combined with (A.5) yields that

$$\sup_{0 \leq t \leq T} \mathbb{E}_x \left[|f_{\mathrm{NN}}(x; \rho_t) - f_{\mathrm{NN}}(x; \tilde{\rho}_t)|^2\right] \leq \left(W_2^2(\rho_0, \tilde{\rho}_0) + \epsilon\right) C_f e^{C_\Delta T}.$$

Then we can conclude (3.8) be setting $\epsilon \to 0$ and $C_s = C_f e^{C_\Delta T}$. $\square$

**Corollary A.3.** *Under the same setting as in Theorem 3.6, one has*

$$\sup_{0 \leq t \leq T} |\mathcal{E}(\rho_t) - \mathcal{E}(\tilde{\rho}_t)| \leq \left(C_s \mathbb{E}_z \left[|h^*(z)|^2\right]\right)^{1/2} W_2(\rho_0, \tilde{\rho}_0) + \frac{1}{2} C_s W_2^2(\rho_0, \tilde{\rho}_0). \tag{A.7}$$

*Proof.* It can be computed that

$$\mathcal{E}(\rho_t) = \frac{1}{2} \mathbb{E}_x \left[|(f_{\mathrm{NN}}(x; \tilde{\rho}_t) - f^*(x)) + (f_{\mathrm{NN}}(x; \rho_t) - f_{\mathrm{NN}}(x; \tilde{\rho}_t))|^2\right]$$

$$= \mathcal{E}(\tilde{\rho}_t) + \mathbb{E}_x \left[(f_{\mathrm{NN}}(x; \tilde{\rho}_t) - f^*(x))(f_{\mathrm{NN}}(x; \rho_t) - f_{\mathrm{NN}}(x; \tilde{\rho}_t))\right]$$

$$+ \frac{1}{2} \mathbb{E}_x \left[|f_{\mathrm{NN}}(x; \rho_t) - f_{\mathrm{NN}}(x; \tilde{\rho}_t)|^2\right],$$

which implies that

$$\sup_{0 \leq t \leq T} |\mathcal{E}(\rho_t) - \mathcal{E}(\tilde{\rho}_t)| \leq \mathbb{E}_x \left[ |f_{\mathrm{NN}}(x; \tilde{\rho}_t) - f^*(x)|^2 \right]^{1/2} \mathbb{E}_x \left[ |f_{\mathrm{NN}}(x; \rho_t) - f_{\mathrm{NN}}(x; \tilde{\rho}_t)|^2 \right]^{1/2}$$

$$+ \frac{1}{2} \mathbb{E}_x \left[ |f_{\mathrm{NN}}(x; \rho_t) - f_{\mathrm{NN}}(x; \tilde{\rho}_t)|^2 \right]$$

$$\leq \left( C_s \mathbb{E}_{x_V} \left[ |h^*(x_V)|^2 \right] \right)^{1/2} W_2(\rho_0, \tilde{\rho}_0) + \frac{1}{2} C_s W_2^2(\rho_0, \tilde{\rho}_0),$$

where we used Theorem 3.6 and Remark 3.2. $\qquad\square$

*Proof of Lemma A.2.* We first prove (A.5). It can be computed that

$$|f_{\mathrm{NN}}(x; \rho_t) - f_{\mathrm{NN}}(x; \tilde{\rho}_t)| = \left| \int a\sigma(w^\top x) \rho_t(d\theta) - \int \tilde{a}\sigma(\tilde{w}^\top x) \tilde{\rho}_t(d\tilde{\theta}) \right|$$

$$\leq \left| \iint (a - \tilde{a}) \sigma(w^\top x) \gamma_t(d\theta, d\tilde{\theta}) \right| + \left| \iint \tilde{a} \left( \sigma(w^\top x) - \sigma(\tilde{w}^\top x) \right) \gamma_t(d\theta, d\tilde{\theta}) \right|,$$

and hence that

$$\mathbb{E}_x \left[ |f_{\mathrm{NN}}(x; \rho_t) - f_{\mathrm{NN}}(x; \tilde{\rho}_t)|^2 \right]$$

$$\leq 2\mathbb{E}_x \left[ \left| \iint (a - \tilde{a}) \sigma(w^\top x) \gamma_t(d\theta, d\tilde{\theta}) \right|^2 \right] + 2\mathbb{E}_x \left[ \left| \iint \tilde{a} \left( \sigma(w^\top x) - \sigma(\tilde{w}^\top x) \right) \gamma_t(d\theta, d\tilde{\theta}) \right|^2 \right].$$

We then bound the two terms above as follows:

$$\mathbb{E}_x \left[ \left| \iint (a - \tilde{a}) \sigma(w^\top x) \gamma_t(d\theta, d\tilde{\theta}) \right|^2 \right] \leq K_\sigma^2 \iint |a - \tilde{a}|^2 \gamma_t(d\theta, d\tilde{\theta}) \leq K_\sigma^2 \Delta(t),$$

and

$$\mathbb{E}_x \left[ \left| \iint \tilde{a} \left( \sigma(w^\top x) - \sigma(\tilde{w}^\top x) \right) \gamma_t(d\theta, d\tilde{\theta}) \right|^2 \right]$$

$$\leq K_\sigma^2 \left( K_\rho + \sqrt{2} K_\xi K_\sigma \mathcal{E}(\rho_0)^{1/2} T \right)^2 \mathbb{E}_x \left[ \left| \iint |(w - \tilde{w})^\top x| \gamma_t(d\theta, d\tilde{\theta}) \right|^2 \right]$$

$$\leq K_\sigma^2 \left( K_\rho + K_\xi K_\sigma \mathbb{E}_z \left[ |h^*(z)|^2 \right]^{1/2} T \right)^2 \iint \mathbb{E}_x \left[ |(w - \tilde{w})^\top x|^2 \right] \gamma_t(d\theta, d\tilde{\theta})$$

$$= K_\sigma^2 \left( K_\rho + K_\xi K_\sigma \mathbb{E}_z \left[ |h^*(z)|^2 \right]^{1/2} T \right)^2 \iint \|w - \tilde{w}\|^2 \gamma_t(d\theta, d\tilde{\theta})$$

$$\leq K_\sigma^2 \left( K_\rho + K_\xi K_\sigma \mathbb{E}_z \left[ |h^*(z)|^2 \right]^{1/2} T \right)^2 \Delta(t),$$

where we used Lemma A.1 and $(w - \tilde{w})^\top x \sim \mathcal{N}(0, \|w - \tilde{w}\|^2)$ if $x \sim \mathcal{N}(0, I_d)$. Then we can conclude (A.5) with $C_f = K_\sigma^2 + K_\sigma^2 \left( K_\rho + K_\xi K_\sigma \mathbb{E}_z \left[ |h^*(z)|^2 \right]^{1/2} T \right)^2$ by combining all estimations above.

We then head into the proof of (A.6), for which we need the particle dynamics

$$\begin{cases} \frac{d}{dt} a_t = \xi_a(t) \mathbb{E}_x \left[ (f_{\mathrm{NN}}(x; \rho_t) - f^*(x)) \sigma(w_t^\top x) \right], \\ \frac{d}{dt} \tilde{a}_t = \xi_a(t) \mathbb{E}_x \left[ (f_{\mathrm{NN}}(x; \tilde{\rho}_t) - f^*(x)) \sigma(\tilde{w}_t^\top x) \right], \end{cases}$$

and

$$\begin{cases} \frac{d}{dt} w_t = \xi_w a_t \mathbb{E}_x \left[ (f_{\mathrm{NN}}(x; \rho_t) - f^*(x)) \sigma'(w_t^\top x) x \right], \\ \frac{d}{dt} \tilde{w}_t = \xi_w \tilde{a}_t \mathbb{E}_x \left[ (f_{\mathrm{NN}}(x; \tilde{\rho}_t) - f^*(x)) \sigma'(\tilde{w}_t^\top x) x \right]. \end{cases}$$

The distance between the particle dynamics can be decomposed as

$$\left| \frac{d}{dt} a_t - \frac{d}{dt} \tilde{a}_t \right| \leq K_\xi \left| \mathbb{E}_x \left[ (f_{\mathrm{NN}}(x; \rho_t) - f_{\mathrm{NN}}(x; \tilde{\rho}_t)) \sigma(w_t^\top x) \right] \right|$$

$$+ K_\xi \left| \mathbb{E}_x \left[ (f_{\mathrm{NN}}(x; \tilde{\rho}_t) - f^*(x))(\sigma(w_t^\top x) - \sigma(\tilde{w}_t^\top x)) \right] \right|, \tag{A.8}$$

and

$$\left|\left\langle w_t - \tilde{w}_t, \frac{d}{dt}w_t - \frac{d}{dt}\tilde{w}_t \right\rangle\right|$$

$$\leq K_\xi \left|(a_t - \tilde{a}_t)\mathbb{E}_x\left[(f_{\mathrm{NN}}(x;\rho_t) - f^*(x))\sigma'(w_t^\top x)(w_t - \tilde{w}_t)^\top x\right]\right| \tag{A.9}$$

$$+ K_\xi \left|\tilde{a}_t \mathbb{E}_x\left[(f_{\mathrm{NN}}(x;\rho_t) - f_{\mathrm{NN}}(x;\tilde{\rho}_t))\sigma'(w_t^\top x)(w_t - \tilde{w}_t)^\top x\right]\right|$$

$$+ K_\xi \left|\tilde{a}_t \mathbb{E}_x\left[(f_{\mathrm{NN}}(x;\tilde{\rho}_t) - f^*(x))\left(\sigma'(w_t^\top x) - \sigma'(\tilde{w}_t^\top x)\right)(w_t - \tilde{w}_t)^\top x\right]\right|.$$

Therefore, it can be computed that

$$\frac{d}{dt}\iint |a - \tilde{a}|^2 \gamma_t(d\theta, d\tilde{\theta})$$

$$= \frac{d}{dt}\iint |a_t - \tilde{a}_t|^2 \gamma_0(d\theta_0, d\tilde{\theta}_0)$$

$$= 2\iint (a_t - \tilde{a}_t)\left(\frac{d}{dt}a_t - \frac{d}{dt}\tilde{a}_t\right)\gamma_0(d\theta_0, d\tilde{\theta}_0)$$

$$\leq \iint |a_t - \tilde{a}_t|^2 \gamma_0(d\theta_0, d\tilde{\theta}_0) + \iint \left|\frac{d}{dt}a_t - \frac{d}{dt}\tilde{a}_t\right|^2 \gamma_0(d\theta_0, d\tilde{\theta}_0)$$

$$\overset{\text{(A.8)}}{\leq} \Delta(t) + 2K_\xi^2 K_\sigma^2 \iint \mathbb{E}_x\left[|f_{\mathrm{NN}}(x;\rho_t) - f_{\mathrm{NN}}(x;\tilde{\rho}_t)|\right]^2 \gamma_0(d\theta_0, d\tilde{\theta}_0)$$

$$+ 2K_\xi^2 K_\sigma^2 \iint \mathbb{E}_x\left[|f_{\mathrm{NN}}(x;\tilde{\rho}_t) - f^*(x)| \cdot \left|(w_t - \tilde{w}_t)^\top x\right|\right]^2 \gamma_0(d\theta_0, d\tilde{\theta}_0)$$

$$\overset{\text{(A.5)}}{\leq} \Delta(t) + 2K_\xi^2 K_\sigma^2 C_f \Delta(t)$$

$$+ 2K_\xi^2 K_\sigma^2 \iint \mathbb{E}_x\left[|f_{\mathrm{NN}}(x;\tilde{\rho}_t) - f^*(x)| \cdot \left|(w_t - \tilde{w}_t)^\top x\right|\right]^2 \gamma_0(d\theta_0, d\tilde{\theta}_0)$$

$$\leq \Delta(t) + 2K_\xi^2 K_\sigma^2 C_f \Delta(t)$$

$$+ 2K_\xi^2 K_\sigma^2 \iint \mathbb{E}_x\left[|f_{\mathrm{NN}}(x;\tilde{\rho}_t) - f^*(x)|^2\right]\mathbb{E}_x\left[\left|(w_t - \tilde{w}_t)^\top x\right|^2\right]\gamma_0(d\theta_0, d\tilde{\theta}_0)$$

$$\leq \Delta(t) + 2K_\xi^2 K_\sigma^2 C_f \Delta(t) + 2K_\xi^2 K_\sigma^2 \mathbb{E}_z[|h^*(z)|^2]\iint \|w_t - \tilde{w}_t\|^2 \gamma_0(d\theta_0, d\tilde{\theta}_0)$$

$$\leq \left(1 + 2K_\xi^2 K_\sigma^2 C_f + 2K_\xi^2 K_\sigma^2 \mathbb{E}_z[|h^*(z)|^2]\right)\Delta(t),$$

where we used Remark 3.2, and that

$$\frac{d}{dt}\iint \|w - \tilde{w}\|^2 \gamma_t(d\theta, d\tilde{\theta})$$

$$= \frac{d}{dt}\iint \|w_t - \tilde{w}_t\|^2 \gamma_0(d\theta_0, d\tilde{\theta}_0)$$

$$= 2\iint \left\langle w_t - \tilde{w}_t, \frac{d}{dt}w_t - \frac{d}{dt}\tilde{w}_t \right\rangle \gamma_0(d\theta_0, d\tilde{\theta}_0)$$

$$\overset{\text{(A.9)}}{\leq} 2K_\xi K_\sigma \iint |a_t - \tilde{a}_t| \cdot \mathbb{E}_x\left[|f_{\mathrm{NN}}(x;\rho_t) - f^*(x)| \cdot \left|(w_t - \tilde{w}_t)^\top x\right|\right]\gamma_0(d\theta_0, d\tilde{\theta}_0)$$

$$+ 2K_\xi K_\sigma \int\int |\tilde{a}_t| \cdot \mathbb{E}_x\left[|f_{\mathrm{NN}}(x;\rho_t) - f_{\mathrm{NN}}(x;\tilde{\rho}_t))| \cdot \left|(w_t - \tilde{w}_t)^\top x\right|\right]\gamma_0(d\theta_0, d\tilde{\theta}_0)$$

$$+ 2K_\xi K_\sigma \iint |\tilde{a}_t| \cdot \mathbb{E}_x\left[|f_{\mathrm{NN}}(x;\tilde{\rho}_t) - f^*(x)| \cdot \left|(w_t - \tilde{w}_t)^\top x\right|^2\right]\gamma_0(d\theta_0, d\tilde{\theta}_0)$$

$$\leq 2K_\xi K_\sigma \iint |a_t - \tilde{a}_t| \cdot \mathbb{E}_z\left[|h^*(z)|^2\right]^{1/2}\mathbb{E}_x\left[\left|(w_t - \tilde{w}_t)^\top x\right|^2\right]^{1/2}\gamma_0(d\theta_0, d\tilde{\theta}_0)$$

$$+ 2K_\xi K_\sigma \int\int |\tilde{a}_t| \cdot (C_f\Delta(t))^{1/2}\mathbb{E}_x\left[\left|(w_t - \tilde{w}_t)^\top x\right|^2\right]^{1/2}\gamma_0(d\theta_0, d\tilde{\theta}_0)$$

$$+ 2K_\xi K_\sigma \iint |\tilde{a}_t| \cdot \mathbb{E}_z\left[|h^*(z)|^2\right]^{1/2}\mathbb{E}_x\left[\left|(w_t - \tilde{w}_t)^\top x\right|^4\right]^{1/2}\gamma_0(d\theta_0, d\tilde{\theta}_0)$$

$$\leq 2K_\xi K_\sigma \mathbb{E}_z \left[|h^*(z)|^2\right]^{1/2} \iint |a_t - \tilde{a}_t| \cdot \|w_t - \tilde{w}_t\| \gamma_0(d\theta_0, d\tilde{\theta}_0)$$

$$+ 2K_\xi K_\sigma \left(K_\rho + K_\xi K_\sigma \mathbb{E}_z \left[|h^*(z)|^2\right]^{1/2} T\right) (C_f \Delta(t))^{1/2} \int\int \|w_t - \tilde{w}_t\| \gamma_0(d\theta_0, d\tilde{\theta}_0)$$

$$+ 2K_\xi K_\sigma \left(K_\rho + K_\xi K_\sigma \mathbb{E}_z \left[|h^*(z)|^2\right]^{1/2} T\right) \mathbb{E}_z \left[|h^*(z)|^2\right]^{1/2}$$

$$\cdot \iint \sqrt{3}\|w_t - \tilde{w}_t\|^2 \gamma_0(d\theta_0, d\tilde{\theta}_0)$$

$$\leq K_\xi K_\sigma \mathbb{E}_z \left[|h^*(z)|^2\right]^{1/2} \Delta(t) + 2K_\xi K_\sigma \left(K_\rho + K_\xi K_\sigma \mathbb{E}_z \left[|h^*(z)|^2\right]^{1/2} T\right) C_f^{1/2} \Delta(t)$$

$$+ 2\sqrt{3} K_\xi K_\sigma \left(K_\rho + K_\xi K_\sigma \mathbb{E}_z \left[|h^*(z)|^2\right]^{1/2} T\right) \mathbb{E}_z \left[|h^*(z)|^2\right]^{1/2} \Delta(t),$$

where we used Remark 3.2, (A.5), Lemma A.1, and $(w - \tilde{w})^\top x \sim \mathcal{N}(0, \|w - \tilde{w}\|^2)$ if $x \sim \mathcal{N}(0, I_d)$. Therefore, we can conclude that

$$\frac{d}{dt}\Delta(t) = \frac{d}{dt}\iint |a - \tilde{a}|^2 \gamma_t(d\theta, d\tilde{\theta}) + \frac{d}{dt}\iint \|w - \tilde{w}\|^2 \gamma_t(d\theta, d\tilde{\theta}) \leq C_\Delta \Delta(t),$$

where

$$C_\Delta = 1 + 2K_\xi^2 K_\sigma^2 C_f + 2K_\xi^2 K_\sigma^2 \mathbb{E}_{x_V}[|h^*(x_V)|^2]$$

$$+ K_\xi K_\sigma \mathbb{E}_z \left[|h^*(z)|^2\right]^{1/2} + 2K_\xi K_\sigma \left(K_\rho + K_\xi K_\sigma \mathbb{E}_z \left[|h^*(z)|^2\right]^{1/2} T\right) C_f^{1/2}$$

$$+ 2\sqrt{3} K_\xi K_\sigma \left(K_\rho + K_\xi K_\sigma \mathbb{E}_z \left[|h^*(z)|^2\right]^{1/2} T\right) \mathbb{E}_z \left[|h^*(z)|^2\right]^{1/2}.$$

This proves (A.6). $\qquad\square$

### A.3 Proof of Theorem 3.4

The proof of Theorem 3.4 is based on Theorem 3.5 and Theorem 3.6.

*Proof of Theorem 3.4.* Let $\mathcal{P}_S : \mathbb{R}^{d+1} \to S$ be the projection in Theorem 3.5 and let $\mathcal{P}_S^\perp = I_{d+1} - \mathcal{P}_S$. Let $\tilde{\rho}_t$ solve (2.4) with $\tilde{\rho}_0 = \mathcal{P}_S^\perp \rho_w \times \delta_S$. It is clear that $(\mathcal{P}_S)_\# \tilde{\rho}_0 = \delta_S$. In addition, with the decomposition $x = x_1 + x_2 + x_3$ where $x_1 = x_V^\perp = x - x_V$, $x_2 = x_V - x_S$, and $x_3 = x_S$ are independent Gaussian random variables, we have for any $w \in \mathbb{R}^d$ that

$$\mathbb{E}_x \left[f^*(x)\sigma'\left(w^\top x_S^\perp\right) x_S\right] = \mathbb{E}_{x_1}\mathbb{E}_{x_2,x_3}\left[\left[h^*(x_2+x_3)\sigma'\left(w^\top x_1 + w^\top x_2\right) x_3\right]\right] = 0,$$

where we used (3.1). Then according to Theorem 3.5, for any $t \geq 0$ that $(\mathcal{P}_S)_\# \tilde{\rho}_t = \delta_S$, which implies that $f_{\mathrm{NN}}(x; \rho_t)$ is a constant function in $x_S$ for any fixed $x_S^\perp$, giving a lowerbound on its loss:

$$\mathcal{E}(\tilde{\rho}_t) = \frac{1}{2}\mathbb{E}_x \left[\|f^*(x) - f_{\mathrm{NN}}(x; \tilde{\rho}_t)\|^2\right]$$

$$= \frac{1}{2}\mathbb{E}_{x_S^\perp} \left[\mathbb{E}_{x_S} \left[\|f^*(x) - f_{\mathrm{NN}}(x; \tilde{\rho}_t)\|^2\right]\right]$$

$$\geq \frac{1}{2}\mathbb{E}_{x_S^\perp} \left[\mathbb{E}_{x_S} \left[\|f^*(x) - \mathbb{E}_{x_S}[f^*(x)]\|^2\right]\right]$$

$$= \frac{1}{2}\mathbb{E}_{z_S^\perp} \left[\mathbb{E}_{z_S} \left[\|h^*(z) - h_{S^\perp}^*(z_S^\perp)\|^2\right]\right]$$

$$= \frac{1}{2}\mathbb{E}_z \left[\|h^*(z) - h_{S^\perp}^*(z_S^\perp)\|^2\right],$$

where $z_S = \mathcal{P}_S^V z$, $z_S^\perp = z - z_S$, and $h_{S^\perp}^*(z_S^\perp) = \mathbb{E}_{z_S}[h^*(z)]$.

Now we show the actual flow is not very different. For $\rho_0 = \rho_a \times \rho_w$ with $\rho_w \sim \mathcal{N}(0, I_d)$ and $\tilde{\rho}_0 = \mathcal{P}_S^\perp \rho_w \times \delta_S$, it can be estimated that

$$W_2^2(\rho_0, \tilde{\rho}_0) \leq \frac{\dim S}{d} \leq \frac{p}{d}.$$

Then applying Corollary A.3, we can conclude for any $T > 0$ that

$$\inf_{0 \le t \le T} \mathcal{E}(\tilde{\rho}_t) \ge \inf_{0 \le t \le T} \mathcal{E}(\tilde{\rho}_t) - \left(C_s \mathbb{E}_z \left[|h^*(z)|^2\right]\right)^{1/2} W_2(\rho_0, \tilde{\rho}_0) - \frac{1}{2} C_s W_2^2(\rho_0, \tilde{\rho}_0)$$

$$\ge \frac{1}{2} \mathbb{E}_z \left[\|h^*(z) - h_{S^\perp}^*(z_S^\perp)\|^2\right] - \frac{\left(p C_s \mathbb{E}_z \left[|h^*(z)|^2\right]\right)^{1/2}}{d^{1/2}} - \frac{p C_s}{2d},$$

where $C_s > 0$ is the constant in Theorem 3.6 depending only on $p$, $h^*$, $K_\sigma$, $K_\xi$, $K_\rho$, and $T$. Therefore, we can obtain (3.3) and (3.4) immediately. $\qquad\square$

# B  Further Discussion and Characterization of the Reflective Property and Theorem 3.4

## B.1  Equivalence between (3.2) and isoLeap$(h^*) \ge 2$

We prove the following equivalence where isoLeap$(h^*)$ is the isotropic leap complexity defined in [2, Appendix B.2].

**Proposition B.1.** *For any polynomial $h^* : V \to \mathbb{R}$, then it satisfies (3.2) with some nontrivial subspace $S \subset V$ if and only if* isoLeap$(h^*) \ge 2$.

*Proof.* Without loss of generality, we assume that $V = \mathbb{R}^p$. Suppose that isoLeap$(h^*) \ge 2$, which means that the leap complexity of $h^*$ (as defined in [2, Definition 1]) is greater than one for some orthonormal basis of $V$. We can assume that the basis is $\{e_1, e_2, \ldots, e_p\}$, where $e_j$ is the vector in $\mathbb{R}^p$ with the $j$-th entry being 1 and other entries being 0. Denote the Hermite decomposition of $h^*$ as

$$h^*(z) = \sum_{i=1}^m c_i \prod_{j=1}^p \mathrm{He}_{\alpha_i(j)}(z_j), \tag{B.1}$$

where $c_1, c_2, \ldots, c_m$ are nonzero coefficients, $\alpha_1, \alpha_2, \ldots, \alpha_m$ are pairwise distinct elements in $\mathbb{N}^p$ with $\alpha_i(j)$ being the $j$-th entry of $\alpha_i$, and $\mathrm{He}_k$ is the $k$-th order Hermite polynomial. Since the leap complexity of $h^*$ is at least two, the following is true after applying some permutation on $\{1, 2, \ldots, m\}$: There exists some $m_1 \in \{1, 2, \ldots, m - 1\}$, such that it holds for any $i \in \{m_1 + 1, \ldots, m\}$ that

$$\sum_{j \in J_{m_1}} \alpha_i(j) \ge 2, \tag{B.2}$$

where

$$J_{m_1} = \{j \in \{1, 2, \ldots, p\} : \alpha_i(j) = 0, \, \forall\, i \in \{1, 2, \ldots, m_1\}\}.$$

Thus, by the orthogonality of Hermite polynomials, we have for $i \in \{m_1 + 1, \ldots, m\}$ that

$$\mathbb{E}_{z_j \sim \mathcal{N}(0,1)} \left[ z_j \prod_{j=1}^p \mathrm{He}_{\alpha_i(j)}(z_j) \right] = 0, \quad \forall\, j \in J_{m_1}.$$

The above also holds for $i \in \{1, 2, \ldots, m_1\}$ by the definition of $J_{m_1}$. Therefore, we can conclude that

$$\mathbb{E}_{z_S \sim \mathcal{N}(0, I_S)}[h^*(z) z_S] = 0, \quad \forall\, z_S^\perp,$$

i.e., (3.2) holds, for $S = \mathrm{span}\{e_j : j \in J_{m_1}\}$. Moreover, $S$ is nontrivial since (B.2) implies $J_{m_1}$ is not empty.

On the other hand, suppose that (3.2) is satisfied with some nontrivial subspace $S \subset V$ that can be assumed as $S = \{(z_1, \ldots, z_{p_1}, 0, \ldots, 0) : z_1, \ldots, z_{p_1} \in \mathbb{R}\}$ with $1 \le p_1 \le p$. We still consider the Hermite decomposition as in (B.1) and rewrite it as

$$h^*(z) = \sum_{i=1}^{m_2} c_i' \prod_{j=1}^{p_1} \mathrm{He}_{\alpha_i'(j)}(z_j) h_i(z_{p_1+1}, \ldots, z_p),$$

where $c_1', c_2', \ldots, c_{m_2}'$ are nonzero coefficients, $\alpha_1', \alpha_2', \ldots, \alpha_{m_2}'$ are pairwise distinct elements in $\mathbb{N}^{p_1}$, and $h_1, h_2, \ldots, h_{m_2}$ are nonzero polynomials defined on $\mathbb{R}^{p-p'}$. Then it follows from (3.2) that

$$\sum_{i=1}^{m_2} c_i' h_i(z_{p_1+1}, \ldots, z_p) \mathbb{E}_{z_j \sim \mathcal{N}(0,1)} \left[ z_j \mathrm{He}_{\alpha_i'(j)}(z_j) \right] \prod_{j'=1, j' \neq j}^{p_1} \mathbb{E}_{z_{j'} \sim \mathcal{N}(0,1)} \mathrm{He}_{\alpha_i'(j')}(z_{j'}) = 0,$$

for any $j \in \{1, 2, \ldots, p_1\}$ and $z_{p_1+1}, \ldots, z_p \in \mathbb{R}$, which implies that

$$\sum_{j=1}^{p_1} \alpha_i(j)' \neq 1, \quad \forall\, i \in \{1, 2, \ldots, m_2\}.$$

Therefore, the leap complexity of $h^*$ is at least 2 with respect to this basis, which leads to isoLeap$(h^*) \geq 2$. $\qquad\qquad\square$

## B.2 Discretization Results Implied by Theorem 3.4

We discuss the sample complexity result of SGD implied by Theorem 3.4 in this subsection. Recall that there have been standard dimension-free results for bounding the distance between SGD and the mean-field dynamics; see e.g., [19]. So the result in this subsection is somehow a direct corollary. However, one needs to make minor modifications to guarantee that all boundedness assumptions in [19] are satisfied.

Given a constant $C_f^b > 0$, define

$$\tilde{f}^*(x) = \mathrm{sign}(f^*(x)) \min\{|f^*(x), C_f^b|\}, \tag{B.3}$$

which is bounded with $|\tilde{f}^*(x)| \leq C_f^b$. One observation is that the subspace-sparse structure of $f^*$ implies that for any $\delta > 0$, there exists a dimension-free constant $C_f^b$ depending on $h^*$ and $\delta$ such that $\mathbb{E}_{x \sim \mathcal{N}(0, I_d)}[|f^*(x) - \tilde{f}^*(x)|^2] < \delta$. The associated mean-field dynamics is

$$\begin{cases} \partial_t \tilde{\rho}_t = \nabla_\theta \cdot \left( \tilde{\rho}_t \xi(t) \nabla_\theta \tilde{\Phi}(\theta; \tilde{\rho}_t) \right), \\ \tilde{\rho}_t \big|_{t=0} = \rho_0, \end{cases} \tag{B.4}$$

where the learning rate $\xi(t) = \mathrm{diag}(\xi_a(t), \xi_w(t) I_d)$ and the initialization $\rho_0$ are shared with (2.4), and

$$\tilde{\Phi}(\theta; \rho) = a \mathbb{E}_{x \sim \mathcal{N}(0, I_d)} \left[ \left( f_{\mathrm{NN}}(x; \rho) - \tilde{f}^*(x) \right) \sigma(w^\top x) \right].$$

The corresponding SGD is given by

$$\theta_i^{(k+1)} = \theta_i^{(k)} + \gamma^{(k)} \left( \tilde{f}^*(x_k) - f_{\mathrm{NN}}(x_k; \Theta^{(k)}) \right) \nabla_\theta \tau(x_k; \theta_i^{(k)}), \quad i = 1, 2, \ldots, N, \tag{B.5}$$

where $N$ is the number of neurons and $\gamma^{(k)} = \mathrm{diag}(\gamma_a^{(k)}, \gamma_w^{(k)} I_d) \succeq 0$ is the learning rate with $\gamma_a^{(k)} = \epsilon \xi_a(k\epsilon)$ and $\gamma_w^{(k)} = \epsilon \xi_w(k\epsilon)$.

Suppose that assumptions made in Theorem 3.4 hold and fix $T > 0$. Using similar analysis as in Appendix A.2, one can conclude that for any $\delta > 0$, there exists a dimension-free constant $C_f^b$ such that

$$\sup_{0 \leq t \leq T} |\mathcal{E}(\rho_t) - \mathcal{E}(\tilde{\rho}_t)| < \delta.$$

Applying Theorem 3.4 and [19, Theorem 1], we can conclude that for any $\mu \in (0, 1)$, there exists dimension-free constants $N_0, d_0, C_\epsilon$, such that for any $N \geq N_0$ and $d \geq d_0$, the following holds with probability at least $\mu$ for any $\epsilon \leq \frac{C_\epsilon}{d + \log N}$:

$$\inf_{k \in [0, T/\epsilon] \cap \mathbb{N}} \mathcal{E}_N(\Theta^{(k)}) \geq \frac{1}{8} \mathbb{E}_{z \sim \mathcal{N}(0, I_V)} \left[ |h^*(z) - h_{S^\perp}^*(z_S^\perp)|^2 \right] > 0.$$

If we further assume that $N = \mathcal{O}(e^d)$, this indicates that SGD as in (B.5) cannot learn the subspace-sparse polynomial $f^*$ within finite time horizon and with $\mathcal{O}(d)$ samples/data points.

## C   Proofs for Section 4

### C.1   Approximation of $w(a, t)$ by Polynomials

This subsection follows [1] closely to approximate and analyze the behavior of $w(a, t)$ for $0 \le t \le T$ with $\xi_a(t) = 0$ and $\xi_w(t) = 1$. The dynamics of a single particle starting at $\theta = (a, 0) \in \mathbb{R}^{d+1}$ can be described by the following ODE:

$$\begin{cases} \frac{\partial}{\partial t} w(a, t) = a \mathbb{E}_x \left[ g(x, t) \sigma'(w(a, t)^\top x) x \right], \\ w(a, 0) = 0, \end{cases} \tag{C.1}$$

where

$$g(x, t) = f^*(x) - f_{\text{NN}}(x; \rho_t)$$

is the residual. The first observation is that

$$\mathbb{E}_x \left[ f^*(x) \sigma(w^\top x_V) x_V^\perp \right] = \mathbb{E}_{x_V} \left[ h^*(x_V) \sigma(w^\top x_V) \mathbb{E}_{x_V^\perp} \left[ x_V^\perp \right] \right] = 0, \quad \forall w \in \mathbb{R}^d.$$

By Theorem 3.5, we have that $\rho_t$ is supported in $\left\{ (a, w) : w_V^\perp = 0 \right\}$, and hence that

$$w_V^\perp(a, t) = 0, \quad \forall 0 \le t \le T.$$

We then analyze the behaviour of $w_V(a, t)$. Let $\{e_1, e_2, \ldots, e_p\}$ be an orthonormal basis of $V$ and we denote $w_i = w^\top e_i$ and $x_i = x^\top e_i$ for any $w, x \in \mathbb{R}^d$ and $i \in \{1, 2, \ldots, p\}$.

By Assumption 4.2, it holds that

$$\sigma'\left(w(a, t)^\top x\right) = m_1 + \sum_{l=1}^{L-1} \frac{m_{l+1}}{l!} (w(a, t)^\top x)^l + \mathcal{O}\left((w(a, t)^\top x)^L\right) \tag{C.2}$$

with $m_l = \sigma^{(l)}(0)$ and then an approximated solution to (C.1) (by polynomial expansion with high-order terms omitted) can be written as

$$\tilde{w}_i(a, t) = \sum_{1 \le j \le L} Q_{i,j}(t) a^j, \quad 1 \le i \le p,$$

where $Q(t)$ is given by $Q(0) = 0$ and the following dynamics:

$$\begin{cases} \frac{d}{dt} Q_{i,1}(t) = \mathbb{E}_x[x_i g(x, t) m_1], \\ \frac{d}{dt} Q_{i,j}(t) = \mathbb{E}_x \left[ x_i g(x, t) \sum_{l=1}^{L-1} \frac{m_{l+1}}{l!} \sum_{1 \le i_1, \ldots, i_l \le p} \sum_{j_1 + \cdots + j_l = j-1} \prod_{s=1}^{l} Q_{i_s, j_s}(t) x_{i_s} \right], \quad 2 \le j \le L. \end{cases} \tag{C.3}$$

Let us remark that even if every single $\tilde{w}_i(a, t)$ depends on the basis $\{e_1, e_2, \ldots, e_p\}$, the linear combination

$$\tilde{w}(a, t) = \sum_{i=1}^{p} e_i \tilde{w}_i(a, t) \tag{C.4}$$

is basis-independent. To see this, let $Q_j(t) \in \mathbb{R}^d$ be the $j$-th column of $Q(t)$ and it holds that

$$\begin{cases} \frac{d}{dt} Q_1(t) = \mathbb{E}_x[x g(x, t) m_1], \\ \frac{d}{dt} Q_j(t) = \mathbb{E}_x \left[ x g(x, t) \sum_{l=1}^{L-1} \frac{m_{l+1}}{l!} \sum_{j_1 + \cdots + j_l = j-1} \prod_{s=1}^{l} x^\top Q_{j_s}(t) \right], \end{cases}$$

The distance between $w_i(a, t)$ and $\tilde{w}_i(a, t)$ for $i \in I$ can be bounded as follows.

**Proposition C.1.** *Suppose that Assumption 4.2 holds. Then*

$$|w_i(a, t) - \tilde{w}_i(a, t)| = \mathcal{O}((|a|t)^{L+1}), \quad \forall i \in \{1, 2, \ldots, p\}. \tag{C.5}$$

We need the following two lemmas to prove Proposition C.1.

**Lemma C.2.** *For any $i \in \{1, 2 \ldots, p\}$ and $j \in \{1, 2, \ldots, L\}$, it holds that*

$$Q_{i,j}(t) = \mathcal{O}(t^j). \tag{C.6}$$

*Proof.* The non-increasing property of the energy functional implies that

$$2\mathcal{E}(\rho_t) = \mathbb{E}_x[|g(x,t)|^2] \leq \mathbb{E}_x[|f^*(x) - f_{\mathrm{NN}}(x, \rho_0)|^2] = \mathbb{E}_z[|h^*(z)|^2] < +\infty.$$

Then (C.6) can be proved by induction. For $j = 1$, it follows from the boundedness of

$$\left| \frac{d}{dt} Q_{i,1}(t) \right| = |\mathbb{E}_x[x_i g(x,t) m_1]| \leq |m_1| \left( \mathbb{E}_x[x_i^2] \cdot \mathbb{E}_x[|g(x,t)|^2] \right)^{1/2}$$

that $Q_{i,e_l}(t) = \mathcal{O}(t)$. Consider any $2 \leq j \leq L$ and assume that $Q_{i,j'} = \mathcal{O}(t^{j'})$ holds for any $1 \leq i \leq p$ and $1 \leq j' < j$. Then one has that

$$\left| \frac{d}{dt} Q_{i,j}(t) \right|$$

$$\leq \left( \mathbb{E}_x[|g(x,t)|^2] \cdot \mathbb{E}_x \left[ \left| x_i \sum_{l=1}^{L-1} \frac{m_{l+1}}{l!} \sum_{1 \leq i_1, \ldots, i_l \leq p} \sum_{j_1 + \cdots + j_l = j-1} \prod_{s=1}^{l} Q_{i_s, j_s}(t) x_{i_s} \right|^2 \right] \right)^{1/2}$$

$$= \mathcal{O}(t^{j-1}),$$

which implies that $Q_{i,j}(t) = \mathcal{O}(t^j)$. $\qquad \square$

**Lemma C.3.** *Suppose that Assumption 4.2 holds. We have for all $i \in \{1, 2, \ldots, p\}$ that*

$$w_i(a, t) = \mathcal{O}(|a|t). \tag{C.7}$$

*Proof.* There exists a constant $C > 0$ and a open subset $A \subset \{w \in \mathbb{R}^d : w_V^{\perp} = 0\}$ containing 0, such that

$$\left| \mathbb{E}_x \left[ x_i g(x,t) \sigma'(w^{\top} x) \right] \right| \leq C, \quad \forall w \in A, \ t \geq 0, \ i \in \{1, 2, \ldots, p\}.$$

Thus, we have

$$\left| \frac{\partial}{\partial t} w_i(a, t) \right| \leq |a|,$$

as long as $w(a, t)$ does not leave $A$. This implies (C.7). $\qquad \square$

Now we can proceed to prove Proposition C.1.

*Proof of Proposition C.1.* Set $\tilde{w}_V^{\perp}(a, t) = 0$. It can be estimated for any $i \in \{1, 2, \ldots, p\}$ that

$$\left| \frac{\partial}{\partial t} \tilde{w}_i(a, t) - a \mathbb{E}_x \left[ x_i g(x,t) \left( m_1 + \sum_{l=1}^{L-1} \frac{m_{l+1}}{l!} (\tilde{w}(a,t)^{\top} x)^l \right) \right] \right|$$

$$\leq \left| \sum_{1 \leq j \leq L} \frac{d}{dt} Q_{i,j}(t) a^j - a \mathbb{E}_x \left[ x_i g(x,t) \left( m_1 + \sum_{l=1}^{L-1} \frac{m_{l+1}}{l!} \left( \sum_{1 \leq i' \leq p} \sum_{1 \leq j \leq L} Q_{i',j}(t) a^j x_{i'} \right)^l \right) \right] \right|$$

$$\leq \sum_{L+1 \leq j \leq L^{L-1}+1} \left| a^j \cdot \mathbb{E}_x \left[ x_i g(x,t) \sum_{l=1}^{L-1} \frac{m_{l+1}}{l!} \sum_{1 \leq i_1, \ldots, i_l \leq p} \sum_{j_1 + \cdots + j_l = j-1} \prod_{s=1}^{l} Q_{i_s, j_s}(t) x_{i_s} \right] \right|$$

$$\leq \sum_{L+1 \leq j \leq L^{L-1}+1} |a|^j$$

$$\cdot \left( \mathbb{E}[|g(x,t)|^2] \cdot \mathbb{E}_x \left[ \left| x_i \sum_{l=1}^{L-1} \frac{m_{l+1}}{l!} \sum_{1 \leq i_1, \ldots, i_l \leq p} \sum_{j_1 + \cdots + j_l = j-1} \prod_{s=1}^{l} Q_{i_s, j_s}(t) x_{i_s} \right|^2 \right] \right)^{1/2}$$

$$=\mathcal{O}(|a|^{L+1}t^L),$$

which combined with (C.2) yields that

$$\frac{\partial}{\partial t}\sum_{i=1}^{p}|w_i(a,t)-\tilde{w}_i(a,t)|$$

$$\leq\sum_{i=1}^{p}\left|\frac{\partial}{\partial t}w_i(a,t)-\frac{\partial}{\partial t}\tilde{w}_i(a,t)\right|$$

$$\leq\sum_{i=1}^{p}\left|a\mathbb{E}_x\left[x_i g(x,t)\left(m_1+\sum_{l=1}^{L-1}\frac{m_{l+1}}{l!}(w(a,t)^\top x)^l\right)\right]\right.$$

$$\left.-a\mathbb{E}_x\left[x_i g(x,t)\left(m_1+\sum_{l=1}^{L-1}\frac{m_{l+1}}{l!}(\tilde{w}(a,t)^\top x)^l\right)\right]\right|$$

$$+|a\mathbb{E}_x[x_i g(x,t)]|\cdot\mathcal{O}\left((w(a,t)^\top x)^L\right)+\mathcal{O}\left(|a|^{L+1}t^L\right)$$

$$\leq\sum_{i=1}^{p}\left|\mathbb{E}_x\left[x_i a^\top g(x,t)\sum_{l=1}^{L-1}\frac{m_{l+1}}{l!}\left((w(a,t)^\top x)^l-(\tilde{w}(a,t)^\top x)^l\right)\right]\right|+\mathcal{O}\left(|a|^{L+1}t^L\right)$$

$$=\mathcal{O}\left(\sum_{i=1}^{p}|w_i(a,t)-\tilde{w}_i(a,t)|\right)+\mathcal{O}\left(|a|^{L+1}t^L\right).$$

Then one can conclude (C.5) from Gronwall's inequality. $\qquad\square$

Even if $\tilde{w}(a,t)$ approximates $w(a,t)$ using polynomial expansion, the coefficients $Q(t)$ are still very difficult to analyze. Thus, we follow [1] to consider the following dynamics that is obtained by replacing $g(x,t)=f^*(x)-f_{\mathrm{NN}}(x;\rho_t)$ by $f^*(x)$ in (C.3):

$$\hat{w}_i(a,t)=\sum_{1\leq j\leq L}\hat{Q}_{i,j}(t)a^j,\quad i\in\{1,2,\ldots,p\},\tag{C.8}$$

where $\hat{Q}(t)$ is given by $\hat{Q}(0)=0$ and

$$\begin{cases}\frac{d}{dt}\hat{Q}_{i,1}(t)=\mathbb{E}_x[x_i f^*(x)m_1],\\\frac{d}{dt}\hat{Q}_{i,j}(t)=\mathbb{E}_x\left[x_i f^*(x)\sum_{l=1}^{L-1}\frac{m_{l+1}}{l!}\sum_{1\leq i_1,\ldots,i_l\leq p}\sum_{j_1+\cdots+j_l=j-1}\prod_{s=1}^{l}\hat{Q}_{i_s,j_s}(t)x_{i_s}\right],\quad 2\leq j\leq L.\end{cases}\tag{C.9}$$

Similar to $\tilde{w}(a,t)$, the linear combination $\hat{w}(a,t)$ defined as

$$\hat{w}(a,t)=\sum_{i=1}^{p}e_i\hat{w}_i(a,t)$$

is also independent of the orthogonal basis $\{e_1,e_2,\ldots,e_p\}$. $\hat{Q}(t)$ can be understood clearly as follows.

**Proposition C.4.** *For any $i\in\{1,2,\ldots,p\}$ and $j\in\{1,2,\ldots,L\}$, there exists a constant $\hat{q}_{i,j}$ depending only on $h^*$ and $\sigma$, such that*

$$\hat{Q}_{i,j}(t)=\hat{q}_{i,j}t^j.\tag{C.10}$$

*Proof.* The proof is straightforward by induction on $j$. $\qquad\square$

The next proposition quantifies the distance between $\hat{Q}(t)$ and $Q(t)$.

**Proposition C.5.** *It holds for any $i\in\{1,2,\ldots,p\}$ and $j\in\{1,2,\ldots,L\}$ that*

$$|Q_{i,j}(t)-\hat{Q}_{i,j}(t)|=\mathcal{O}(t^{j+1}).\tag{C.11}$$

We need the following lemma for the proof of Proposition C.5.

**Lemma C.6.** *It holds that*

$$\left(\mathbb{E}_x\left[|f_{NN}(x;\rho_t)|^2\right]\right)^{1/2} = \mathcal{O}(t). \tag{C.12}$$

*Proof.* Noticing that $f_{\mathrm{NN}}(x;\rho_0) = 0$ by the symmetry of $\rho_a$, one has that

$$
\begin{aligned}
f_{\mathrm{NN}}(x;\rho_t) &= f_{\mathrm{NN}}(x;\rho_t) - f_{\mathrm{NN}}(x;\rho_0) \\
&= \int a\left(\sigma(w(a,t)^\top x) - \sigma(0)\right)\rho_a(da) \\
&= \int a\left(\sum_{l=1}^L \frac{m_l}{l!}(w(a,t)^\top x)^l + \mathcal{O}\left((w(a,t)^\top x)^{L+1}\right)\right)\rho_a(da),
\end{aligned}
$$

and hence by Lemma C.3 and $w_V^\perp(a,t) = 0$ that

$$\mathbb{E}_x\left[|f_{\mathrm{NN}}(x,\rho_t)|^2\right] = \mathcal{O}(t^2),$$

which implies (C.12). $\qquad\square$

*Proof of Proposition C.5.* We prove (C.11) by introduction on $j$. For $j = 1$, one has

$$\left|\frac{d}{dt}Q_{i,1}(t) - \frac{d}{dt}\hat{Q}_{i,1}(t)\right| = |\mathbb{E}_x[x_i f_{\mathrm{NN}}(x,\rho_t)m_1]| = |m_1|\left(\mathbb{E}_x[x_i^2]\cdot\mathbb{E}_x[|f_{\mathrm{NN}}(x,\rho_t)|^2]\right)^{1/2} = \mathcal{O}(t),$$

which leads to $|Q_{i,e_l}(t) - \hat{Q}_{i,e_l}(t)| = \mathcal{O}(t^2)$. Then we consider $2 \leq j \leq L$ and assume that $|Q_{i,j'}(t) - \hat{Q}_{i,j'}(t)| = \mathcal{O}(t^{j'+1})$ holds for $1 \leq j' < j$. It can be estimated that

$$
\begin{aligned}
&\left|\frac{d}{dt}Q_{i,j}(t) - \frac{d}{dt}\hat{Q}_{i,j}(t)\right| \\
=&\left|\mathbb{E}_x\left[x_i g(x,t)\sum_{l=1}^{L-1}\frac{m_{l+1}}{l!}\sum_{1\leq i_1,\dots,i_l\leq p}\sum_{j_1+\cdots+j_l=j-1}\prod_{s=1}^l Q_{i_s,j_s}(t)x_{i_s}\right]\right. \\
&\left. -\mathbb{E}_x\left[x_i f^*(x)\sum_{l=1}^{L-1}\frac{m_{l+1}}{l!}\sum_{1\leq i_1,\dots,i_l\leq p}\sum_{j_1+\cdots+j_l=j-1}\prod_{s=1}^l \hat{Q}_{i_s,j_s}(t)x_{i_s}\right]\right| \\
=&\left|\mathbb{E}_x\left[x_i f_{\mathrm{NN}}(x;\rho_t)\sum_{l=1}^{L-1}\frac{m_{l+1}}{l!}\sum_{1\leq i_1,\dots,i_l\leq p}\sum_{j_1+\cdots+j_l=j-1}\prod_{s=1}^l Q_{i_s,j_s}(t)x_{i_s}\right]\right. \\
&\left. -\mathbb{E}_x\left[x_i f^*(x)\sum_{l=1}^{L-1}\frac{m_{l+1}}{l!}\sum_{1\leq i_1,\dots,i_l\leq p}\sum_{j_1+\cdots+j_l=j-1}\left(\prod_{s=1}^l Q_{i_s,j_s}(t) - \prod_{s=1}^l \hat{Q}_{i_s,j_s}(t)\right)x_{i_s}\right]\right| \\
=&\left|\mathbb{E}_x\left[x_i f_{\mathrm{NN}}(x,\rho_t)\sum_{l=1}^{L-1}\frac{m_{l+1}}{l!}\sum_{1\leq i_1,\dots,i_l\leq p}\sum_{j_1+\cdots+j_l=j-1}\prod_{s=1}^l \mathcal{O}(t^{j_s})x_{i_s}\right]\right. \\
&\left. - \mathbb{E}_x\left[x_i f^*(x)\sum_{l=1}^{L-1}\frac{m_{l+1}}{l!}\sum_{1\leq i_1,\dots,i_l\leq p}\sum_{j_1+\cdots+j_l=j-1}\right.\right. \\
&\left.\left.\qquad\qquad\qquad\left(\prod_{s=1}^l(\hat{q}_{i_s,j_s}t^{j_s} + \mathcal{O}(t^{j_s+1})) - \prod_{s=1}^l \hat{q}_{i_s,j_s}t^{j_s}\right)x_{i_s}\right]\right| \\
=&\mathcal{O}(t^j),
\end{aligned}
$$

where we used Lemma C.6. Then one can conclude (C.11). $\qquad\square$

## C.2 From Linear Independence to Algebraic Independence

We prove Theorem 4.5 in this subsection.

**Definition C.7** (Algebraic independence). *Let $v_1, v_2, \ldots, v_m \in \mathbb{R}[a_1, a_2, \ldots, a_p]$ be polynomials in $a_1, a_2, \ldots, a_p$. We say that $v_1, v_2, \ldots, v_m$ are algebraically independent if for any nonzero polynomial $F : \mathbb{R}^m \to \mathbb{R}$,*

$$F(v_1(a_1, a_2, \ldots, a_p), \ldots, v_m(a_1, a_2, \ldots, a_p)) \neq 0 \in \mathbb{R}[a_1, a_2, \ldots, a_p].$$

**Lemma C.8.** *If $v_1, v_2, \ldots, v_p \in \mathbb{R}[a]$ are $\mathbb{R}$-linearly independent, then*

$$\det \begin{pmatrix} v_1(a_1) & v_1(a_2) & \cdots & v_1(a_p) \\ v_2(a_1) & v_2(a_2) & \cdots & v_2(a_p) \\ \vdots & \vdots & \ddots & \vdots \\ v_p(a_1) & v_p(a_2) & \cdots & v_p(a_p) \end{pmatrix} \tag{C.13}$$

*is a non-zero polynomial in $\mathbb{R}[a_1, a_2, \ldots, a_p]$.*

*Proof.* Let $n_i$ be the smallest degree of nonzero monomials of $v_i$ and let $c_i$ be the accociated coefficient for $i = 1, 2, \ldots, p$. Without loss of generality, we assume that $n_0 < n_1 < \cdots < n_p$ (otherwise one can perform some row reductions or row permutations). The polynomial defined in (C.13) consists of monomials of degree at least $n_1 + n_2 + \cdots + n_p$. So it suffices to prove that the sum of monomials with degree being $n_1 + n_2 + \cdots + n_p$ is nonzero, which is true since

$$\det \begin{pmatrix} c_1 a_1^{n_1} & c_1 a_2^{n_1} & \cdots & c_1 a_p^{n_1} \\ c_2 a_1^{n_2} & c_2 a_2^{n_2} & \cdots & c_2 a_p^{n_2} \\ \vdots & \vdots & \ddots & \vdots \\ c_p a_1^{n_p} & c_p a_2^{n_p} & \cdots & c_p a_p^{n_p} \end{pmatrix} = c_1 c_2 \ldots c_p \cdot \det \begin{pmatrix} a_1^{n_1} & a_2^{n_1} & \cdots & a_p^{n_1} \\ a_1^{n_2} & a_2^{n_2} & \cdots & a_p^{n_2} \\ \vdots & \vdots & \ddots & \vdots \\ a_1^{n_p} & a_2^{n_p} & \cdots & a_p^{n_p} \end{pmatrix}$$

is nonzero as a generalized Vandermonde matrix. $\qquad\square$

*Proof of Theorem 4.5.* Since $v_1, v_2, \ldots, v_p \in \mathbb{R}[a]$ be $\mathbb{R}$-linearly independent with the constant terms being zero, we can see that $v_1', v_2', \ldots, v_p' \in \mathbb{R}[a]$ are also $\mathbb{R}$-linearly independent. Noticing that

$$\frac{\partial}{\partial a_j} \left( \frac{1}{p}(v_i(a_1) + v_i(a_2) + \cdots + v_i(a_p)) \right) = \frac{1}{p} v_i'(a_j),$$

one can conclude that $\frac{1}{p}(v_1(a_1) + \cdots + v_1(a_p)), \ldots, \frac{1}{p}(v_p(a_1) + \cdots + v_p(a_p)) \in \mathbb{R}[a_1, \ldots, a_p]$ are $\mathbb{R}$-algebraically independent by using Theorem 4.6 and Lemma C.8. $\qquad\square$

## C.3 Proofs of Proposition 4.4 and Theorem 4.3

Some ideas in this subsection are from [1], but the proofs are significantly different since we need to show the algebraic independence to obtain a non-degenerate kernel, as discussed in Section 4.

**Lemma C.9.** *Suppose that Assumption 4.1 holds with $s \in \mathbb{N}_+$. There exists some orthonormal basis $\{e_1, e_2, \ldots, e_p\}$ of $V$ such that the coefficients $\hat{q}_{i,j}, 1 \leq i \leq p, 1 \leq j \leq L$ in (C.10) satisfies*

$$s_1 < s_2 < \cdots < s_p \leq s, \tag{C.14}$$

*where*

$$s_i = \min\{j : \hat{q}_{i,j} \neq 0\}, \quad i = 1, 2, \ldots, p.$$

*Proof.* Let $\hat{w}_V(t)$ be the dynamics defined in (4.1). It can be seen that the Taylor's expansion of $\hat{w}_V(t)$ at $t = 0$ up to $s$-th order is given by

$$\sum_{i=1}^{p} \sum_{j=1}^{s} e_i \hat{Q}_{i,j}(t) = \sum_{i=1}^{p} \sum_{j=1}^{s} e_i \hat{q}_{i,j} t^j,$$

where $\hat{Q}_{i,j}$ and $\hat{q}_{i,j}$ are as in (C.9) and (C.5). According to Assumption 4.1, the matrix $(\hat{q}_{i,j})_{1 \leq i \leq p, 1 \leq j \leq s}$ is of full-row-rank. One can thus perform the QR decomposition, or equivalently choose some orthogonal basis, to obtain (C.14). $\qquad\square$

In the rest of this subsection, we will always denote $\mathbf{s} = (s_1, s_2, \ldots, s_P)$ and

$$u(a_1, \ldots, a_p, t) = \frac{1}{p}(w(a_1, t) + w(a_2, t) + \cdots + w(a_p, t)),$$

$$\tilde{u}(a_1, \ldots, a_p, t) = \frac{1}{p}(\tilde{w}(a_1, t) + \tilde{w}(a_2, t) + \cdots + \tilde{w}(a_p, t)),$$

$$\hat{u}(a_1, \ldots, a_p, t) = \frac{1}{p}(\hat{w}(a_1, t) + \hat{w}(a_2, t) + \cdots + \hat{w}(a_p, t)),$$

for $t \in [0, T]$, where $w(a, t)$, $\tilde{w}(a, t)$, and $\hat{w}(a, t)$ are defined in (C.1), (C.4), and (C.8), respectively. Recall that $p_1, p_2, \ldots, p_{\binom{n+p}{p}}$ are the orthonormal basis of $\mathbb{P}_{V,n}$ with input $z \sim \mathcal{N}(0, I_V)$, where $\mathbb{P}_{V,n}$ is the collection of all polynomials on $V$ with degree at most $n = \deg(h^*) = \deg(f^*)$. Proposition 4.4 aims to bound from below the smallest eigenvalue of the kernel matrix (4.2) whose definition is restated as follows

$$\mathcal{K}_{i_1, i_2}(t) = \mathbb{E}_{a_1, \ldots, a_p} \left[ \mathbb{E}_{z, z'} \left[ p_{i_1}(z) \hat{\sigma}(u(a_1, \ldots, a_p, t)^\top z) \hat{\sigma}(u(a_1, \ldots, a_p, t)^\top z') p_{i_2}(z') \right] \right],$$

where $\hat{\sigma}(\xi) = (1 + \xi)^n$, $(a_1, \ldots, a_p) \sim \mathcal{U}([-1, 1]^p)$, and $1 \leq i_1, i_2 \leq \binom{n+p}{p}$. To do this, we define three $\binom{n+p}{p} \times \binom{n+p}{p}$ matrices

$$\tilde{\mathcal{K}}_{i_1, i_2}(t) = \mathbb{E}_{a_1, \ldots, a_p} \left[ \mathbb{E}_{z, z'} \left[ p_{i_1}(z) \hat{\sigma}(\tilde{u}(a_1, \ldots, a_p, t)^\top z) \hat{\sigma}(\tilde{u}(a_1, \ldots, a_p, t)^\top z') p_{i_2}(z') \right] \right],$$

and

$$\tilde{M}_{i_1, i_2}(\mathbf{a}, t) = \mathbb{E}_z \left[ p_{i_1}(z) \hat{\sigma}(\tilde{u}(\mathbf{a}_{i_2}, t)^\top z) \right],$$

$$\hat{M}_{i_1, i_2}(\mathbf{a}, t) = \mathbb{E}_z \left[ p_{i_1}(z) \hat{\sigma}(\hat{u}(\mathbf{a}_{i_2}, t)^\top z) \right],$$

where $\mathbf{a} = \left( \mathbf{a}_1, \mathbf{a}_2, \ldots, \mathbf{a}_{\binom{n+p}{p}} \right)$ and $\mathbf{a}_i \in \mathbb{R}^p$ for $i = 1, 2, \ldots, \binom{n+p}{p}$.

**Lemma C.10.** *It holds that*

$$\det(\hat{M}(\mathbf{a}, t)) = \sum_{i=1}^{\binom{n+p}{p}} \sum_{0 \leq \|\mathbf{j}_i\|_1 \leq Ln} \hat{h}_{\mathbf{j}} t^{\|\mathbf{j}\|_1} \mathbf{a}^{\mathbf{j}}, \tag{C.15}$$

*where $\mathbf{j} = \left( \mathbf{j}_1, \mathbf{j}_2, \ldots, \mathbf{j}_{\binom{n+p}{p}} \right)$ with $\mathbf{j}_i \in \mathbb{N}^p$ for $i = 1, 2, \ldots, \binom{n+p}{p}$, $\hat{h}_{\mathbf{j}}$ is a constant depending on $h^*$ and $\sigma$, and $\mathbf{a}^{\mathbf{j}}$ represents the product of entrywise powers.*

*Proof.* The result follows directly from Proposition C.4. $\qquad\square$

**Lemma C.11.** *It holds that*

$$\det(\tilde{M}(\mathbf{a}, t)) = \sum_{i=1}^{\binom{n+p}{p}} \sum_{0 \leq \|\mathbf{j}_i\|_1 \leq Ln} \tilde{h}_{\mathbf{j}}(t) \mathbf{a}^{\mathbf{j}},$$

*with*

$$\tilde{h}_{\mathbf{j}}(t) = \hat{h}_{\mathbf{j}} t^{\|\mathbf{j}\|_1} + \mathcal{O}(t^{\|\mathbf{j}\|_1 + 1}).$$

*Proof.* The result follows directly from Proposition C.4 and Proposition C.5. $\qquad\square$

**Lemma C.12.** *For any $m \in \mathbb{N}$, one has that*

$$\text{span} \left\{ (q^\top z)^m : q \in V \right\} = \mathbb{P}_{V,m}^h, \tag{C.16}$$

*where $\mathbb{P}_{V,m}^h$ is the collection of all homogeneous polynomials in $z \in V$ with degree $m$.*

*Proof.* Without loss of generality, we assume that $V = \mathbb{R}^p$ and prove the result by induction on $p$. (C.16) is clearly true for $p = 1$. Then we consider $p \geq 2$ and assume that (C.16) holds for $p - 1$ and any $m$.

For any $q, z \in \mathbb{R}^p$, we denote that $\bar{q} = (q_2, \ldots, q_p)$ and $\bar{z} = (z_2, \ldots, z_p)$. Let $t_0, t_1, \ldots, t_m \in \mathbb{R}$ be distinct. Then it follows from the invertibility of $(t_i^j)_{0 \le i,j \le m}$ and

$$
\begin{pmatrix} (t_0 z_1 + \bar{q}^\top \bar{z})^m \\ (t_1 z_1 + \bar{q}^\top \bar{z})^m \\ \vdots \\ (t_n z_1 + \bar{q}^\top \bar{z})^m \end{pmatrix} = \begin{pmatrix} \sum_{i=0}^m \binom{m}{i} t_0^i z_1^i (\bar{q}^\top \bar{z})^{m-i} \\ \sum_{i=0}^m \binom{m}{i} t_1^i z_1^i (\bar{q}^\top \bar{z})^{m-i} \\ \vdots \\ \sum_{i=0}^m \binom{m}{i} t_m^i z_1^i (\bar{q}^\top \bar{z})^{m-i} \end{pmatrix}
$$

$$
= \begin{pmatrix} 1 & t_0 & \cdots & t_0^m \\ 1 & t_1 & \cdots & t_1^m \\ \vdots & \vdots & \ddots & \vdots \\ 1 & t_m & \cdots & t_m^m \end{pmatrix} \begin{pmatrix} \binom{m}{0}(\bar{q}^\top \bar{z})^m \\ \binom{m}{1} z_1 (\bar{q}^\top \bar{z})^{m-1} \\ \vdots \\ \binom{m}{m} z_1^m \end{pmatrix}
$$

that

$$
z_1^i (\bar{q}^\top \bar{z})^{m-i} \in \text{span} \left\{ (r^\top z)^m : r \in \mathbb{R}^p \right\}, \quad \forall \bar{q} \in \mathbb{R}^{p-1}, \ i \in \{0, 1, \ldots, m\}.
$$

Then using the induction hypothesis that (C.16) is true for $p-1$ and $m-i$, $i = 0, 1, \ldots, m$, one can conclude that (C.16) is also true for $p$ and $m$. $\qquad \square$

**Lemma C.13.** *For $\hat{\sigma}(\xi) = (1 + \xi)^n$, one has that*

$$
\text{span} \left\{ \hat{\sigma}(q^\top z) : q \in V \right\} = \mathbb{P}_{V,n}.
$$

*Proof.* One can still assume that $V = \mathbb{R}^p$ without loss of generality. Consider any $q \in \mathbb{R}^p$ and any distinct $t_0, t_1, \ldots, t_n \in \mathbb{R}$. Then

$$
\begin{pmatrix} \hat{\sigma}((t_0 q)^\top z) \\ \hat{\sigma}((t_1 q)^\top z) \\ \vdots \\ \hat{\sigma}((t_n q)^\top z) \end{pmatrix} = \begin{pmatrix} \sum_{i=0}^n \binom{n}{i} t_0^i (q^\top z)^i \\ \sum_{i=0}^n \binom{n}{i} t_1^i (q^\top z)^i \\ \vdots \\ \sum_{i=0}^n \binom{n}{i} t_0^i (q^\top z)^i \end{pmatrix} = \begin{pmatrix} 1 & t_0 & \cdots & t_0^n \\ 1 & t_1 & \cdots & t_1^n \\ \vdots & \vdots & \ddots & \vdots \\ 1 & t_n & \cdots & t_n^n \end{pmatrix} \begin{pmatrix} \binom{n}{0} \\ \binom{n}{1} q^\top z \\ \vdots \\ \binom{n}{n} (q^\top z)^n \end{pmatrix}.
$$

Note that the matrix $(t_i^j)_{0 \le i,j \le n}$ is invertible when $t_0, t_1, \ldots, t_N$ are distinct. Therefore, one can conclude that

$$
(q^\top z)^i \in \text{span} \left\{ \sigma(r^\top z) : r \in \mathbb{R}^p \right\}, \quad \forall q \in \mathbb{R}^p, \ 0 \le i \le n,
$$

which combined with Lemma C.12 implies that

$$
\mathbb{P}_{V,n} \supset \text{span} \left\{ \hat{\sigma}(q^\top z) : q \in \mathbb{R}^p \right\} \supset \mathbb{P}_{V,0}^h \oplus \mathbb{P}_{V,1}^h \oplus \cdots \oplus \mathbb{P}_{V,n}^h = \mathbb{P}_{V,n},
$$

which completes the proof. $\qquad \square$

*Proof of Lemma 4.7.* We prove the result by induction. When $m = 1$, the result follows directly from the $\mathbb{R}$-algebraic independence of $v_1, v_2, \ldots, v_p$. Now we assume that the result is true for $m-1$ and consider the case of $m$. Suppose that

$$
0 = \sum_{\mathbf{j}_1, \ldots, \mathbf{j}_m} X_{\mathbf{j}} \prod_{l=1}^m \mathbf{v}(\mathbf{a}_l)^{\mathbf{j}_l} = \sum_{\mathbf{j}_m} \left( \sum_{\mathbf{j}_1 \ldots, \mathbf{j}_{m-1}} X_{\mathbf{j}} \prod_{l=1}^{m-1} \mathbf{v}(\mathbf{a}_l)^{\mathbf{j}_l} \right) \mathbf{v}(\mathbf{a}_m)^{\mathbf{j}_m}.
$$

By the $\mathbb{R}$-algebraic independence of $v_1, v_2, \ldots, v_p$, one must have

$$
\sum_{\mathbf{j}_1 \ldots, \mathbf{j}_{m-1}} X_{\mathbf{j}} \prod_{l=1}^{m-1} \mathbf{v}(\mathbf{a}_l)^{\mathbf{j}_l} = 0, \quad \forall \mathbf{j}_m,
$$

which then leads to $X_{\mathbf{j}} = 0$, $\forall \mathbf{j}$ by the induction hypothesis. $\qquad \square$

**Lemma C.14.** *Suppose that Assumption 4.1 and 4.2 hold and let $\hat{h}_{\mathbf{j}}$ be the coefficient of $\det(\hat{M}(\mathbf{a}, t))$ in (C.15). Then there exists some $\hat{\mathbf{j}} = \left( \hat{\mathbf{j}}_1, \hat{\mathbf{j}}_2, \ldots, \hat{\mathbf{j}}_{\binom{n+p}{p}} \right)$ with $\|\hat{\mathbf{j}}\|_1 \le sn\binom{n+p}{p}$, such that $\hat{h}_{\hat{\mathbf{j}}} \ne 0$.*

*Proof.* Let $X(\mathbf{q}) \in \mathbb{R}^{\binom{n+p}{p} \times \binom{n+p}{p}}$ be defined as

$$X_{i_1,i_2}(\mathbf{q}) = \mathbb{E}_z \left[ p_{i_1}(z) \sigma(\mathbf{q}_{i_2}^\top z) \right],$$

where $\mathbf{q} = \left( \mathbf{q}_1, \mathbf{q}_2, \ldots, \mathbf{q}_{\binom{n+p}{p}} \right)$ with $\mathbf{q}_i \in V$. Then $\det(X(\mathbf{q}))$ is a polynomial in $\mathbf{q}$ of the form

$$\det(X(\mathbf{q})) = \sum_{i=1}^{\binom{n+p}{p}} \sum_{0 \le \|\mathbf{j}_i\|_1 \le n} X_{\mathbf{j}} \mathbf{q}^{\mathbf{j}}, \tag{C.17}$$

where we understand $\mathbf{q}_i$ as a (coefficient) vector in $\mathbb{R}^p$ associated with a fixed orthonormal basis $\{e_1, e_2, \ldots, e_p\}$ of $V$ that satisfies Lemma C.9. By Lemma C.13, there exists some $\mathbf{q}$ such that $\sigma(\mathbf{q}_1^\top z), \sigma(\mathbf{q}_2^\top z), \ldots, \sigma(\mathbf{q}_{\binom{n+p}{p}}^\top z)$ form a basis of $\mathbb{P}_{V,n}$, which implies that $\det(X(\mathbf{q})) \neq 0$ for this $\mathbf{q}$. Thus, (C.17) is a non-zero polynomial in $\mathbf{q}$. Let $\mathbf{s} = (s_1, s_2, \ldots, s_p)$ collect all indices $s_i$ from Lemma C.9. Denote

$$S = \min \left\{ \sum_{l=1}^{\binom{n+p}{p}} \mathbf{s}^\top \mathbf{j}_l : X_{\mathbf{j}} \neq 0, \ 0 \le \|\mathbf{j}_i\| \le n, \ 1 \le i \le \binom{n+p}{p} \right\},$$

and

$$J_S = \left\{ \mathbf{j} : \sum_{l=1}^{\binom{n+p}{p}} \mathbf{s}^\top \mathbf{j}_l = S, \ X_{\mathbf{j}} \neq 0, \ 0 \le \|\mathbf{j}_i\| \le n, \ 1 \le i \le \binom{n+p}{p} \right\}.$$

Then we have that

$$\det(\hat{M}(\mathbf{a}, t)) = \det \left( X \left( \hat{u}(\mathbf{a}_1, t), \hat{u}(\mathbf{a}_2, t), \ldots, \hat{u}(\mathbf{a}_{\binom{n+p}{p}}, t) \right) \right)$$

$$= \sum_{i=1}^{\binom{n+p}{p}} \sum_{0 \le \|\mathbf{j}_i\|_1 \le n} X_{\mathbf{j}} \prod_{l=1}^{\binom{n+p}{p}} \hat{u}(\mathbf{a}_l, t)^{\mathbf{j}_l}$$

$$= \sum_{\mathbf{j} \in J_S} X_{\mathbf{j}} \prod_{l=1}^{\binom{n+p}{p}} \hat{u}_{\mathbf{s}}(\mathbf{a}_l, t)^{\mathbf{j}_l} + \mathcal{O}(t^{S+1}),$$

where $\hat{u}_{\mathbf{s}}(\mathbf{a}_l, t) = (\hat{u}_{2,s_1}(\mathbf{a}_l, t), \ldots, \hat{u}_{p,s_p}(\mathbf{a}_l, t))$ and $\hat{u}_{i,s_i}(a_1, \ldots, a_p, t) = \frac{1}{p} \sum_{k=1}^p \hat{q}_{i,s_i} t^{s_i} a_k^{s_i}$ collects the leading order terms of $\hat{u}_i(a_1, \ldots, a_p, t)$. According to Assumption C.9, Theorem 4.5, and Lemma 4.7, we have that

$$\sum_{\mathbf{j} \in J_S} X_{\mathbf{j}} \prod_{l=1}^{\binom{n+p}{p}} \hat{u}_{\mathbf{s}}(\mathbf{a}_l, t)^{\mathbf{j}_l} \neq 0,$$

which provides at least one non-zero term in $\det(\hat{M}(\mathbf{a}, t))$ whose degree in $t$ is

$$S \le sn \binom{n+p}{p}.$$

This completes the proof. □

*Proof of Proposition 4.4.* According to Lemma C.14, there exist some $\hat{\mathbf{j}}$ with $\|\hat{\mathbf{j}}\|_1 \le sn\binom{n+p}{p}$ such that $\hat{h}_{\hat{\mathbf{j}}} \neq 0$. According to Lemma C.11 and Lemma 103 in [1], it holds that

$$\mathbb{E}_{\mathbf{a}} \left[ \left( \det(\tilde{M}(\mathbf{a}, t)) \right)^2 \right] \ge C_1 \left| \tilde{h}_{\hat{\mathbf{j}}}(t) \right|^2 = C_1 \left| \hat{h}_{\hat{\mathbf{j}}} t^{\|\hat{\mathbf{j}}\|_1} + \mathcal{O}(t^{\|\hat{\mathbf{j}}\|_1 + 1}) \right|^2 \ge \frac{C_1}{2} \left| \hat{h}_{\hat{\mathbf{j}}} \right| t^{2\|\hat{\mathbf{j}}\|_1}, \tag{C.18}$$

where $\mathbf{a} \sim \mathcal{U}\left( ([-1,1]^p)^{\binom{n+p}{p}} \right)$ for some constant $C_1$ depending only on $n, p, s$, and for sufficiently small $t$. It can be seen that

$$\tilde{\mathcal{K}}(t) = \frac{1}{\binom{n+p}{p}} \mathbb{E}_{\mathbf{a}} \left[ \tilde{M}(\mathbf{a}, t) \tilde{M}(\mathbf{a}, t)^\top \right].$$

By Jensen's inequality, one has that

$$\lambda_{\min}(\tilde{\mathcal{K}}(t)) \geq \frac{1}{\binom{n+p}{p}} \mathbb{E}_{\mathbf{a}} \left[ \lambda_{\min} \left( \tilde{M}(\mathbf{a}, t) \tilde{M}(\mathbf{a}, t)^\top \right) \right]. \tag{C.19}$$

Since entries of $\tilde{M}(a, t)$ are all $\mathcal{O}(1)$ by Lemma C.2, which implies the boundedness of the eigenvalues $\lambda_i \left( \tilde{M}(\mathbf{a}, t) \tilde{M}(\mathbf{a}, t)^\top \right)$, $i = 1, 2, \ldots, \binom{n+p}{p}$, one has that

$$\left( \det(\tilde{M}(\mathbf{a}, t)) \right)^2 = \det \left( \tilde{M}(\mathbf{a}, t) \tilde{M}(\mathbf{a}, t)^\top \right) \leq C_2 \lambda_{\min} \left( \tilde{M}(\mathbf{a}, t) \tilde{M}(\mathbf{a}, t)^\top \right). \tag{C.20}$$

Combining (C.18), (C.19), and (C.20), one can conclude that

$$\lambda_{\min}(\tilde{\mathcal{K}}(t)) \geq \frac{C_1}{2} \left| \hat{h}_{\hat{\mathbf{j}}}(\alpha, m) \right| t^{2\|\hat{\mathbf{j}}\|_1} \geq \frac{C_1}{2} \left| \hat{h}_{\hat{\mathbf{j}}}(\alpha, m) \right| t^{2sn\binom{n+p}{p}},$$

where we used $\|\hat{\mathbf{j}}\| \leq N\binom{n+p}{p} \max_{i \in I} s_i$ and only considered $t \leq 1$. By Proposition C.1, we have that

$$\left| \lambda_{\min}(\mathcal{K}(t)) - \lambda_{\min}(\tilde{\mathcal{K}}(t)) \right| = \mathcal{O}(t^{L+1}).$$

Then we can obtain (4.3) as we set $L = 2sn\binom{n+p}{p}$. $\qquad\square$

*Proof of Theorem 4.3.* Let $C, T$ be the constants in Proposition 4.4 and consider $t > T$. It follows from Theorem 3.5 that $w_V^\perp(a, T) = 0$, which leads to that $u_V^\perp(a_1, \ldots, a_p, T) = 0$ and hence that $f_{\mathrm{NN}}(x; \rho_t)$ and $g(x, t)$ only depend on $x_V$ for all $t \geq T$. Define $\mathbf{g}(t) \in \mathbb{R}^{\binom{n+p}{p}}$ via $\mathbf{g}_i(t) = \mathbb{E}_x[g(x, t)p_i(x_V)]$. Then according to (2.6) and (4.3), one has for $t > T$ that

$$\begin{aligned}
\frac{d}{dt} \mathcal{E}(\rho_t) &= - \sum_{i=1}^{P} \mathbb{E}_a \mathbb{E}_{x,x'} \left[ g_i(x, t) \sigma(w(a, T)^\top x) \sigma(w(a, T)^\top x') g_i(x', t) \right] \\
&= - \mathbf{g}(t)^\top \mathcal{K}(T) \mathbf{g}(t) \leq -\lambda_{\min}(\mathcal{K}(T)) \|\mathbf{g}(t)\|^2 \\
&= - 2\lambda_{\min}(\mathcal{K}(T)) \mathcal{E}(\rho_t) \leq -2CT^{2sn\binom{n+p}{p}} \mathcal{E}(\rho_t),
\end{aligned}$$

which implies that

$$\mathcal{E}(\rho_t) \leq \mathcal{E}(\rho_T) \exp \left( -2CT^{2sn\binom{n+p}{p}}(t - T) \right),$$

by Gronwall's inequality. Then we obtain the desired exponential decay property by noticing that $\mathcal{E}(\rho_T) = \frac{1}{2} \mathbb{E}_z[\|h^*(z)\|^2]$. $\qquad\square$

# D  Further Discussion and Characterization of Assumption 4.1 and Theorem 4.3

## D.1  Verification of Assumption 4.1

We provide some verification or characterization of Assumption 4.1. Without loss of generality, we fix an orthonormal basis and $V$ and view that $V = \mathbb{R}^p$. The results in this subsection are independent of the choice of the orthonormal basis.

It has been discussed in Section 4.1 that if $\sigma \in \mathcal{C}^s(\mathbb{R})$ with $s = 2^{p-1}$ and $\sigma^{(1)}(0), \sigma^{(2)}(0), \ldots, \sigma^{(p)}(0)$ are all nonzero, then Assumption 4.1 can be verified with $s$ for $h^*(z) = z_1 + z_1 z_2 + \cdots + z_1 z_2 \cdots z_p$, using the calculation in [1, Proposition 33]. Similarly, the same result is also true for $h^*(z) = c_1 z_1 + c_2 z_1 z_2 + \cdots + c_p z_1 z_2 \cdots z_p$ if $c_1, c_2, \ldots, c_p$ are nonzero. More generally, we have the following.

**Proposition D.1.** *Let $\{\alpha_1, \alpha_2, \ldots, \alpha_m\}$ be a set of pairwise distinct elements in $\mathbb{N}^p$ that contains $(1, 0, \ldots, 0), (1, 1, 0, \ldots, 0), \ldots, (1, 1, \ldots, 1)$. If $\sigma \in \mathcal{C}^s(\mathbb{R})$ with $s = 2^{p-1}$ and $\sigma^{(1)}(0), \sigma^{(2)}(0), \ldots, \sigma^{(p)}(0)$ are all nonzero, then*

$$h^*(z) = \sum_{i=1}^{m} c_i \prod_{j=1}^{p} \mathrm{He}_{\alpha_i(j)}(z_j),$$

*satisfies Assumption 4.1 with $s$ unless $(c_1, c_2, \ldots, c_m)$ is in some measure-zero subset of $\mathbb{R}^m$ with respect to the Lebesgue measure.*

*Proof.* We assume that $\alpha_1 = (1, 0, \ldots, 0), \alpha_2 = (1, 1, 0, \ldots, 0), \ldots, \alpha_m = (1, 1, \ldots, 1)$. As in the proof of Lemma C.9, Assumption 4.1 is true with $s$ if and only if the matrix $\hat{q} := (\hat{q}_{i,j})_{1 \leq i \leq p, 1 \leq j \leq s}$ is of full-row-rank, i.e., $\det(\hat{q}\hat{q}^\top) \neq 0$. By (C.9), each entry in $\hat{q}$ is a polynomial in $(c_1, c_2, \ldots, c_m)$, which implies that $\det(\hat{q}\hat{q}^\top)$ is also a polynomial in $(c_1, c_2, \ldots, c_m)$. This polynomial is nonzero since it takes nonzero value at $(1, 1, \ldots, 1, 0, \ldots, 0)$ with the $p$ entries being 1 and all other entries being 0, which is because that Assumption 4.1 is true for $h^*(z) = z_1 + z_1 z_2 + \cdots + z_1 z_2 \cdots z_p$. Finally, the conclusion of Proposition D.1 is true since the set of roots of a nonzero polynomial is of measure zero with respect to the Lebesgue measure. $\square$

### D.2 Discretization Results Implied by Theorem 4.3

The discussion in this subsection is similar to those in Appendix B.2. We slightly modified the flow $\rho_t$ generated by Algorithm 1 to guarantee some boundedness conditions, and then use the standard dimension-free estimate [19] to derive a sample complexity result implied by Theorem 4.3.

We use the same bounded modification $\tilde{f}^*$ as in (B.3) for a given constant $C_f^b > 0$. Similarly, for any $\delta > 0$ and $C_w > 0$, there exists dimension-free constant $C_\sigma^b > 0$ depending on $h^*, \delta, C_w$, such that one can modify the activation function $\hat{\sigma}(\zeta) = (1 + \zeta)^n$ to $\tilde{\sigma}$ satisfying that $\|\tilde{\sigma}\|_{L^\infty(\mathbb{R})} \leq C_\sigma^b, \|\tilde{\sigma}'\|_{L^\infty(\mathbb{R})} \leq C_\sigma^b, \|\tilde{\sigma}''\|_{L^\infty(\mathbb{R})} \leq C_\sigma^b$, and $\mathbb{E}_{x \sim \mathcal{N}(0, I_d)} \left[ |\hat{\sigma}(w^\top x) - \tilde{\sigma}(w^\top x)|^2 \right] < \delta$, $\mathbb{E}_{x \sim \mathcal{N}(0, I_d)} \left[ |\hat{\sigma}(w^\top x) - \tilde{\sigma}(w^\top x)|^2 \right] < \delta$, for all $w \in \mathbb{R}^d$ with $\|w\| \leq C_w$.

The associated mean-field dynamics $\tilde{\rho}_t$, that can be viewed as a slight modification of $\rho_t$ generated by Algorithm 1, is given by (B.4) for $0 \leq t \leq T$ and follows

$$\begin{cases} \partial_t \tilde{\rho}_t = \nabla_\theta \cdot \left( \tilde{\rho}_t \xi(t) \nabla_\theta \tilde{\Phi}'(\theta; \tilde{\rho}_t) \right), \\ \tilde{\rho}_t \big|_{t=T} = \tilde{\rho}_T, \end{cases} \tag{D.1}$$

with

$$\tilde{\Phi}'(\theta; \rho) = a \mathbb{E}_{x \sim \mathcal{N}(0, I_d)} \left[ \left( \tilde{f}_{\text{NN}}(x; \rho) - \tilde{f}^*(x) \right) \tilde{\sigma}(w^\top x) \right],$$

$$\tilde{f}_{\text{NN}}(x; \rho) = \int a \tilde{\sigma}(w^\top x) \rho(da, dw),$$

for $t \geq T$. Here, the learning rate $\xi(t) = \text{diag}(\xi_a(t), \xi_w(t) I_d)$ is shared with Algorithm 1, namely $\xi_a(t) = 0, \xi_w(t) = 1$ for $0 \leq t \leq T$ and $\xi_a(t) = 1, \xi_w(t) = 0$ for $t \geq T$.

The SGD associated to the modified mean-field dynamics is given by

$$\begin{aligned} w_i^{(k+1)} &= w_i^{(k)} + \epsilon \left( \tilde{f}^*(x_k) - f_{\text{NN}}(x_k; \Theta^{(k)}) \right) a_i^{(k)} \sigma' \left( \left( w_i^{(k)} \right)^\top x_k \right) x_k, \\ a_i^{(k+1)} &= a_i^{(k)}, \end{aligned} \tag{D.2}$$

for $i = 1, 2, \ldots, N$ and $k = 0, 1, \ldots, T/\epsilon - 1$, where $N$ is the number of neurons and $T/\epsilon$ is assumed to be an integer, and

$$\begin{aligned} w_i^{(k+1)} &= w_i^{(k)}, \\ a_i^{(k+1)} &= a_i^{(k)} + \epsilon \left( \tilde{f}^*(x_k) - \tilde{f}_{\text{NN}}(x_k; \Theta^{(k)}) \right) \tilde{\sigma} \left( \left( w_i^{(k)} \right)^\top x_k \right), \end{aligned} \tag{D.3}$$

for $i = 1, 2, \ldots, N$ and $k = T/\epsilon, T/\epsilon + 1, \ldots$

Suppose that assumptions made in Theorem 4.3 hold and fix $T' > T > 0$. Using similar analysis as in Appendix A.2, one can conclude that for any $\delta > 0$, there exist dimension-free constants $C_f^b$ and $C_\sigma^b$ such that

$$\sup_{0 \leq t \leq T'} |\mathcal{E}(\rho_t) - \mathcal{E}(\tilde{\rho}_t)| < \delta.$$

Applying Theorem 4.3 and [19, Theorem 1], we can conclude that for any $\mu \in (0, 1)$ and any $\delta > 0$, there exists dimension-free constants $N_0, d_0, C_\epsilon$, such that for any $N \geq N_0$ and $d \geq d_0$, the following holds with probability at least $\mu$ for any $\epsilon \leq \frac{C_\epsilon}{d + \log N}$ with $T/\epsilon \in \mathbb{N}$:

$$\inf_{k \in [0, T'/\epsilon] \cap \mathbb{N}} \mathcal{E}_N(\Theta^{(k)}) < \delta.$$

If we further assume that $N = \mathcal{O}(e^d)$, this indicates that SGD with (D.2) and (D.3) can learn the subspace-sparse polynomial $f^*$ within finite time horizon and with $\mathcal{O}(d)$ samples/data points.

