# OpenReview forum: "Mean-Field Analysis for Learning Subspace-Sparse Polynomials with Gaussian Input"
_NeurIPS.cc/2024/Conference — NeurIPS 2024 poster_

### Official Review · Reviewer_JjNH · 2024-06-28

**Soundness:** 3
**Presentation:** 2
**Contribution:** 2
**Rating:** 5
**Confidence:** 3

**Summary:**

This submission studies the training dynamics of a two-layer neural network in the mean-field limit, when the data is isotropic Gaussian and the target function is a low-dimensional multi-index model. The authors consider the population gradient flow on the squared error, and identify a "reflective property" that is necessary for GD to learn the target function in finite time. A multi-phase gradient-based training algorithm is also proposed to demonstrate that such property is (almost) sufficient for GD learnability. This provides a coordinate-free generalization of the merged staircase property (MSP) in (Abbe et al. 2023).

**Strengths:**

The main contribution of this submission is the introduction of the reflective property of the target function which entails whether the parameter distribution evolved under the mean-field gradient flow can achieve nontrivial overlap with a given subspace in finite time. This "basis-free" extension of MSP to Gaussian data is an interesting result.

**Weaknesses:**

I have reviewed this paper in another venue. The authors addressed some of my concerns in the revision.
I shall reiterate the remaining weaknesses from my previous review.

- This submission only analyzes the mean-field dynamics up to finite time. If $T$ scales with the dimensionality, then it is known that SGD can learn functions with higher leap / information exponent; these functions may not satisfy the reflective property, yet the mean-field dynamics can achieve low population loss. I do not hold this against the authors since this point is already acknowledged in the main text.

- The learning algorithm in Section 4 is non-standard as it requires an averaging over trajectories and a change of activation function during training. Moreover, the authors do not present any learning guarantee for the discretized dynamics, and therefore a sample complexity cannot be inferred from the analysis.

After reading the revised manuscript, my main concern is that the relation between the proposed reflective property and prior definitions of SGD hardness (in the Related Works paragraph starting line 73) is not clearly explained. Specifically, the authors only listed the alternative rotation-invariant conditions on SGD learnability, but did not discuss the advantages of the proposed definition.

- Is the reflective property equivalent to isoLeap > 1, if we do not take the activation $\sigma$ into account (that is, assuming $\sigma$ is sufficiently expressive)? If not, how does it differ from the sufficient conditions in (Abbe et al. 2023) (Bietti et al. 2023)? More explanation is needed.

- The authors brought up the even symmetry condition in (Dandi et al. 2024), which does not exclude high isoLeap functions. In fact, (Lee et al. 2024) (Arnaboldi et al. 2024) have shown that any non-even single-index functions can be learned by variants of online SGD in $T=O(1)$ time horizon (and the discretized dynamics require $n=O(d)$ samples). The authors should clarify if this is consistent with the proposed necessary condition for SGD.
(Lee et al. 2024) *Neural network learns low-dimensional polynomials with SGD near the information-theoretic limit*.
(Arnaboldi et al. 2024) *Repetita iuvant: Data repetition allows sgd to learn high-dimensional multi-index functions*.

**Questions:**

See Weaknesses above.

**Limitations:**

See Weaknesses above.

---

> ### Author Rebuttal · Authors · 2024-08-07
>
> Thanks a lot for your helpful and insightful comments and questions. Please find our responses below:
> * __Connection/difference between the reflective property (equations (3.1)-(3.2)) and $\textup{IsoLeap}(h^*)>1$ (Abbe et al. 2023):__
>     * Our conditions (3.1) and (3.2) are more general than $\textup{IsoLeap}(h^*)>1$ in the sense that (3.1) and (3.2) can be extended straightforwardly to general function $h^*$, while $\textup{IsoLeap}(h^*)$ can only be defined for polynomial $h^*$.
>     * If we only consider a polynomial $h^*$, then it can be prove that $\textup{IsoLeap}(h^*)>1$ implies (3.2). However, we currently do not know whether (3.2) implies $\textup{IsoLeap}(h^*)>1$ or not.
>     * (3.1) is different from (3.2) and $\textup{IsoLeap}(h^*)>1$ when $\sigma$ is less expressive. In our original submission, we give a counter-example with $\sigma(\zeta) = \zeta$, but this is not the only example. There exist other counter-examples with more expressive $\sigma$. In fact, for any $n\geq 2$, consider $h^*(z) = \textup{He}_1(z_1)+\textup{He}_n(z_1) \textup{He}_1(z_2)$ and $\sigma(\zeta) = \zeta^n$. Then it can be verified that (3.1) is satisfied while $\textup{IsoLeap}(h^*)=1$.
> * __Discretized dynamics and sample complexity:__ There have been standard results (see e.g. Mei et al. (2019)) that bound the distance between the SGD and the mean-field dynamics. To apply the results of Mei et al. (2019), one needs the boundedness of $f^*$ and $\sigma$. However, in our work, $f^*$ and the second activation function $\hat{\sigma}$ in Algorithm 1 are bounded, which is the reason that we did not give a sample complexity result in our original submission. Recently, we realized that a sample complexity result can be derived as follows and we will include such analysis in our revision.
>     * Step 1: We slightly modify the mean-field dynamics by replacing $f^*$ and $\hat{\sigma}$ by $\text{sign}(f^*)\min(|f^*|,C_f)$ and $\text{sign}(\hat{\sigma})\min(|\hat{\sigma}|,C_{\sigma})$ that are both bounded. $C_f$ and $C_{\sigma}$ are dimension-free constants guaranteeing that the distance between the original dynamics and the modified dynamics is smaller than $\epsilon$. Let us remark that $C_f$ and $C_{\sigma}$ can be chosen as dimension-free since $f^*$ is subspace-sparse and dynamics in Algorithm 1 does not learn any information about $V^\perp$.
>     * Step 2: Apply the standard results in Mei et al. (2019) to the modified dynamics and then obtain the sample complexity for SGD.
> * __Some explanations for Algorithm 1:__ We agree with you that taking the average in Algorithm 1 is non-standard. The reason is that we want to lift the linear independence to the algebraic independence. For the change of activation function in Algorithm 1, we find it is somehow acceptable in the existing literature (see Abbe et al. (2022)).
> * __Consistency with Lee et al. (2024) and Arnaboldi et al. (2024):__
>     * Lee et al. (2024) require a sample complexity $\mathcal{O}(d\cdot \text{polylog} d)$. So the training time is not constant, due to the $\text{polylog}d$ term. Our work always considers a finite training time $T$ and we definitely agree that SGD may learn a larger family of functions if we allow the training time to scale with the dimension $d$.
>     * Arnaboldi et al. (2024) analyze a variant of SGD, not the original SGD. Our necessary condition considers the mean-field dynamics corresponding to the original SGD.

---

> > ### Comment · Reviewer_JjNH · 2024-08-09
> >
> > > Consistency with Lee et al. (2024) and Arnaboldi et al. (2024). Lee et al. (2024) require a sample complexity $\mathcal{O}(d\cdot \text{polylog} d)$.
> >
> > The authors need to elaborate. Consider single index model $f^*(x) = \text{He}_5(x^\top \theta)$. Does this function satisfy the proposed reflective property?
> > This function is learnable under finite $T$ using the algorithm in Lee et al. (2024). In fact, Dandi et al. (2024) (which the authors cited) showed that two GD steps with batch size $\mathcal{O}(d)$ suffice.

---

> > > ### Author Response · Authors · 2024-08-10
> > >
> > > Dear Reviewer JjNH,
> > >
> > > Thank you very much for your reply and for the insightful comment. The function $f^*(x) = \text{He}_5(x^\top \theta)$ indeed satisfies the reflective property and our results are consistent with Lee et al. (2024) and Dandi et al. (2024):
> > > * If I understand correctly, both Lee et al. (2024) and Dandi et al. (2024) are based on reusing batches. Differently, our work analyzes the mean-field dynamics induced by the original SGD without reusing batches.
> > > * Lee et al. (2024) run the training algorithm for $\Theta(d\cdot\text{polylog}d)$ iterations and the stepsize is chosen as $\eta = \Theta(d^{-1})$. So the training time is $\Theta(d\cdot\text{polylog}d)\cdot \Theta(d^{-1}) = \Theta(\text{polylog}d)$. I would consider this as "almost" finite time, not exactly finite time.
> > >
> > > Please let us know if you have further comments or questions.

---

> > > > ### Comment · Reviewer_JjNH · 2024-08-10
> > > >
> > > > > Lee et al. (2024) run the training algorithm for $\Theta(d\text{polylog}d)$ iterations
> > > >
> > > > For functions that do not satisfy even symmetry, including $\text{He}_5$, finite time is sufficient as seen in Dandi et al. (2024), Lee et al. (2024) and Arnaboldi et al. (2024). This is *not* the reason that the proposed reflective property fails to predict weak recovery.
> > > >
> > > > My point in the original review is that when the authors cite related conditions on SGD learnability, they need to first understand the settings under which these conditions have predictive power, and compare these against the setting studied in this submission.
> > > > As it currently stands, this submission only considers the mean-field *gradient flow* at finite time, and there's no notion of "original SGD" yet. It is expected that if you discretize this flow with online SGD, then the proposed condition is informative. But if you use SGD to minimize the empirical loss, there's a chance that the data-reuse mechanism in Dandi et al. (2024) could appear.

---

> > > > > ### Author Response · Authors · 2024-08-11
> > > > >
> > > > > Dear Reviewer JjNH,
> > > > >
> > > > > Thank you very much for your insightful and valuable comments! We are sorry for the unclarity and we will revise our manuscript according to your suggestions:
> > > > > * We will discuss the discretization of the mean-field gradient flow with online SGD, using standard discretization error bound in Mei et al. (2019). We did not include it in our original submission due to a technical issue from the unboundedness of $f$ and $\hat{\sigma}$. This issue can be resolved with slight modifications, as described in our rebuttal.
> > > > > * We will clearly mention in our revision that our condition and analysis are only for the mean-field gradient and the online SGD (that is the "original SGD" mentioned in our reply) and that the recovery ability of SGD can be improved in some cases when people apply other techniques such as batch reuse, as in Dandi et al. (2024), Lee et al. (2024), and Arnaboldi et al. (2024). We will include the discussion of all these different but related settings and conditions.
> > > > >
> > > > > Please let us know if you have other comments or questions. Thank you again!

---

### Official Review · Reviewer_kar1 · 2024-07-12

**Soundness:** 3
**Presentation:** 3
**Contribution:** 3
**Rating:** 6
**Confidence:** 3

**Summary:**

This manuscript extends previous work on learning sparse polynomials on the hypercube to Gaussian space. To achieve this, the authors introduce a generalization of the merged staircase property studied in previous work. Specifically, they demonstrate that their "reflective property" is necessary to learn polynomials in O(1) time, while a slightly stronger condition, defined in Assumption 4.1, is sufficient for learning.

As is common in the literature, the authors propose a two-stage training process. First, they train the input layer while keeping the output layer fixed. After learning the representation, they fix the input layer and train the output layer to learn the link function.

**Strengths:**

Overall, the work effectively extends the previously studied merged staircase properties to a basis-independent setting. The manuscript is well-written and contributes valuable insights to the multi-index setting, which is an active area of research in the neural network theory community.

**Weaknesses:**

The main weakness of the proposed algorithm is its requirement to run the training algorithm several times to average the weights and change the activation function in the second stage, which differs significantly from practical training algorithms. At the same time, existing work addresses this issue with random feature approximation in the second stage, though this is also not an ideal solution. Therefore, the question of whether we can train input-output layers together remains an open problem in the field.

**Questions:**

The authors did not discuss their "Reflective property" in the context of leap complexity suggested in [1]. It is known that the merged staircase property corresponds to leap 1 functions. In this sense, is there a way to relate the reflective properties with the leap complexity?

Additionally, in the definition of the "Reflective property," they require the condition to hold for any shifted version of the activation (i.e., for any $u \in \mathbb{R}$  in Definition 3.3). However, in Theorem 3.5, they did not use a bias in the activation. What is the reason for this discrepancy?

Furthermore, I am not familiar with the terminology of Dirac delta measure defined on a subspace, and I could not find its definition in the paper either. Could the authors include this definition in the revised version of the paper?

[1] Abbe, E., Boix-Adserà, E., & Misiakiewicz, T. (2022). The merged-staircase property: a necessary and nearly sufficient condition for SGD learning of sparse functions on two-layer neural networks. ArXiv, abs/2202.08658.

**Limitations:**

See the Weaknesses part.

---

> ### Author Rebuttal · Authors · 2024-08-07
>
> Thanks a lot for your helpful and insightful comments and questions. Please find our responses below:
> * __Connection/difference between the reflective property (equations (3.1)-(3.2)) and $\textup{IsoLeap}(h^*)>1$ (Abbe et al. 2023):__
>     * Our conditions (3.1) and (3.2) are more general than $\textup{IsoLeap}(h^*)>1$ in the sense that (3.1) and (3.2) can be extended straightforwardly to general function $h^*$, while $\textup{IsoLeap}(h^*)$ can only be defined for polynomial $h^*$.
>     * If we only consider a polynomial $h^*$, then it can be prove that $\textup{IsoLeap}(h^*)>1$ implies (3.2). However, we currently do not know whether (3.2) implies $\textup{IsoLeap}(h^*)>1$ or not.
>     * (3.1) is different from (3.2) and $\textup{IsoLeap}(h^*)>1$ when $\sigma$ is less expressive. In our original submission, we give a counter-example with $\sigma(\zeta) = \zeta$, but this is not the only example. There exist other counter-examples with more expressive $\sigma$. In fact, for any $n\geq 2$, consider $h^*(z) = \textup{He}_1(z_1)+\textup{He}_n(z_1) \textup{He}_1(z_2)$ and $\sigma(\zeta) = \zeta^n$. Then it can be verified that (3.1) is satisfied while $\textup{IsoLeap}(h^*)=1$.
> * __Definition 3.2 and Theorem 3.5:__ We understand that it is a bit confusing that we use $u+v^\top z_S^\perp$ in Definition 3.2 and use $w^\top x_S^\perp$ without bias term in Theorem 3.5. The reason can be seen in line 479-480, where we write $w^\top x_S^\perp = w^\top x_V^\perp + w^\top (x_V-x_S)$. Then $w^\top x_V^\perp$ is the bias term $u$ and $x_V-x_S$ is the orthogonal complement of $x_S$ in $V$, corresponding to $z_S^\perp$ in Definition 3.2. We will make this point more clear in our revision.
> * __Delta measure on a subspace:__ For a subspace $S$, the delta measure $\delta_S$ is a probability measure on $S$ such that for any continuous and compactly supported function $f:S\to\mathbb{R}$, we have $\int_S f(x)\delta_S(dx) = f(0)$. We will add the definition in our revision.

---

> > ### Comment · Reviewer_kar1 · 2024-08-11
> >
> > I thank the authors for their detailed response.
> >
> > **Connection between isoleap and reflective property:** As Reviewer 7m62 suggests, the isoleap property can be extended to $\ell_2$ integrable functions with respect to the standard Gaussian measure. Moreover, after spending some time on the suggested definition, I am fairly confident that the reflective property is equivalent to isoleap > 1,  if the activation function can be chosen freely in the definition of reflective property. In this case, the major distinction lies in incorporating the Hermite expansion of the activation function, which somewhat weakens the contribution of “proposing a basis-free generalization of the merged-staircase property”.
> >
> > **Definition 3.2 and Theorem 3.5:** Thank you for the clarification.
> >
> > Despite my concern regarding the first point, I would be happy to see this paper accepted at NeurIPS. Therefore, I will keep my score unchanged. However, I strongly urge the authors to address and clarify the issue regarding isoleap in their final submission.

---

> > > ### Author Response · Authors · 2024-08-12
> > >
> > > Dear Reviewer kar1,
> > >
> > > Thank you very much for your valuable comment and for your support. As discussed in the comment of Reviewer 7m62, our equation 3.2 without $\sigma$ is equivalent to IsoLeap$\geq 2$, which we agree with after a careful check. Hence, the contribution on the necessary condition is a condition considering the expressiveness of $\sigma$ and a different analysis. We will clearly discuss the equivalence and modify the description of our contribution accordingly.

---

### Official Review · Reviewer_7m62 · 2024-07-12

**Soundness:** 2
**Presentation:** 3
**Contribution:** 2
**Rating:** 5
**Confidence:** 4

**Summary:**

This paper considers learning a subspace sparse polynomial with Gaussian data, using a two-layer neural network trained in the mean-field regime. Their main results are two fold:
1)They introduce a reflective property that generalizes the non- merged-staircase property from [1] to Gaussian input. They show that this condition is sufficient for the dynamics to be stuck in a suboptimal saddle-subspace for a constant time horizon.
2) They show that a slightly stronger condition is sufficient for some modified dynamics to decay to $0$ exponentially fast.

**Strengths:**

- The paper is reasonably well written, and provides ample details on their technical results.

- The paper extends the results from coordinate sparse on hypercube, to subspace-sparse on Gaussian data (also known as multi-index models in the statistics literature).

- The proof techniques are involved and require careful arguments, in particular for the algebraic independence.

**Weaknesses:**

- I would be careful when claiming novelty for the basis-free generalization of the merged-staircase property: as correctly pointed out in the related works paragraph, several papers [2,7,9,10] have proposed such properties. It should be easy to check that Eq. (3.2) is equivalent to [2,7,9] for leap=1. The difference is that Eq. (3.1) takes into account that the activation might not be expressive enough (see questions).

- Theorem 3.4 seems to be very similar to [1] ([1] goes further and show concentration on a dimensionless dynamics, so the proof does not directly compare to a dynamics that is $0$ on the subspace)

- There are several unnatural simplifications made in the algorithm.
First the dynamics is actually $p$ particles that are trained independently, to avoid dealing with a PDE (then it is just an ODE on a single particle). This does not seem necessary (e.g., [1]). Is there a reason to not directly consider the PDE?
Second, the activation is changed between the two phases of the algorithm.
Third, the result is proved under Assumption 4.1, which looks very much like assuming that the algorithm works (phase 1 of training is roughly this ODE, and therefore it looks like we are assuming that the dynamics converge to prove that the dynamics converge)... This seems to be a major flaw, but could be clarified during discussions. Could a smooth analysis, similar to [1] work here? (see questions)

- The exponential convergence in Theorem 4.3 is confusing. When seeing such a result in the PDE literature, I assume another type of analysis, e.g., log-sobolev inequality. Here, if I am not mistaken, the exponential convergence is simply due to the fact that we switch in the second phase of the algorithm to a linear dynamics, where the kernel has eigenvalues bounded away from 0.

**Questions:**

- Is there no hope to prove Assumption 4.1, even at the price of randomizing some coefficients of $h_*$ or $\sigma’$ and only proving an almost sure result?

- Is definition 3.1 that much different from 3.2? Except in the edge case of $\sigma’$ identically $0$ (which is the only counterexample given), it seems that we can always condition on $z_s^\perp$.

**Limitations:**

Yes

---

> ### Author Rebuttal · Authors · 2024-08-07
>
> Thanks a lot for your helpful and insightful comments and questions. Please find our responses below:
> * __Connection/difference between the reflective property (equations (3.1)-(3.2)) and $\textup{IsoLeap}(h^*)>1$ (Abbe et al. 2023):__
>     * Our conditions (3.1) and (3.2) are more general than $\textup{IsoLeap}(h^*)>1$ in the sense that (3.1) and (3.2) can be extended straightforwardly to general function $h^*$, while $\textup{IsoLeap}(h^*)$ can only be defined for polynomial $h^*$.
>     * If we only consider a polynomial $h^*$, then it can be prove that $\textup{IsoLeap}(h^*)>1$ implies (3.2). However, we currently do not know whether (3.2) implies $\textup{IsoLeap}(h^*)>1$ or not.
>     * (3.1) is different from (3.2) and $\textup{IsoLeap}(h^*)>1$ when $\sigma$ is less expressive. In our original submission, we give a counter-example with $\sigma(\zeta) = \zeta$, but this is not the only example. There exist other counter-examples with more expressive $\sigma$. In fact, for any $n\geq 2$, consider $h^*(z) = \textup{He}_1(z_1)+\textup{He}_n(z_1) \textup{He}_1(z_2)$ and $\sigma(\zeta) = \zeta^n$. Then it can be verified that (3.1) is satisfied while $\textup{IsoLeap}(h^*)=1$.
> * __Connection between Theorem 3.4 and Abbe et al. (2022):__ The goal of Theorem 3.4 is to generalize the result in Abbe et al. (2022) to the rotation-invariant setting. Therefore, the statement of Theorem 3.4 is similar to Abbe et al. (2022). However, the proofs are significantly different since Abbe et al. (2022) analyze the dimension-free dynamics and we do not use that. We directly show that the flow cannot learn any information about $S$ by connecting it with a flow in $\mathbb{R}\times S^\perp$.
> * __Some explanations for Algorithm 1:__
>     * In Algorithm 1, we need to repeat Step 2 for $p$ times, and for each time the flow is an interacting particle system, where the whole system is described by a PDE and the dynamics of a single particle in the interacting system is described by an ODE. Therefore, Step 3 does not train $p$ particles independently, but trains $p$ interacting particle systems. Compared to training only one interacting particle system as in Abbe et al. (2022), we need to repeat this step to lift the linear independence to the algebraic independence.
>     * You are correct that we change the activation function in Step 4. We want to mention that this is somehow acceptable as in existing literature (see Abbe et al. (2022)).
> * __Exponential convergence in Theorem 4.3:__ You are correct that the exponential convergence is from linear dynamics, i.e., the second phase. This idea is standard as in many existing works including Abbe et al. (2022). The most difficult part is to prove that the eigenvalue of the kernel matrix has a positive lower bound.
> * __Verification of Assumption 4.1:__ It is very difficult to directly verify Assumption 4.1 in general. But there are several cases that are manageable and we will let you know if we figure out a more general case during the discussion period.
>     * If $h^*(z) = c_1 z_1+c_2 z_1 z_2 + \cdots + c_p z_1 z_2\cdots z_p$, then Assumption 4.1 can be proved with nonzero coefficients $c_1,c_2,\dots,c_p$.
>     * If $h^*(z) =  \sum_{s\in\mathcal{S}} c(s) \prod_{i=1}^p He_{s_i}(z_i)$, where $\mathcal{S}$ is a finite subset of $\mathbb{Z}_{\geq 0}^p$ with $(1,0\dots,0),(1,1,0,\dots,0),\dots,(1,1,\dots,1)\in\mathcal{S}$, then Assumption 4.1 can be proved in the almost sure sense when $c(s)$ are randomly generated (e.g., from iid Gaussian) for all $s\in\mathcal{S}$.

---

> ### Comment · Reviewer_7m62 · 2024-08-09
>
> I thank the authors for their detailed response and for clearing out some of my confusion. Here are come comments:
>
> - **Concerning isoleap:** The isoleap is defined in terms of the Hermite coefficients in the orthonormal decomposition of the target function, which always exist for squared integrable functions (which is always the case when using squared loss), so it is not limited to polynomials. (The isoleap is always defined and finite for any squared integrable function.)
>
> - **Concerning Assumption (3.2):** Let's consider $S$ to be one direction $z_1$. Then $ h_* (z) = \sum_{k \geq 0} He_k (z_1) h_k (z_{-1}).$ Therefore the condition (3.2) shows that $h_1(z_{-1}) = 0$ for all $z_{-1}$. Hence, in this coordinate basis, $z_1$ only appears in $He_k$ with $k\geq2$, so its leap $\geq 2$ in that basis, and therefore the isoleap is $\geq 2$ (this generalizes straighforwardly to $S$ of any dimension). Hence Assumption (3.2) is equivalent to isoleap $\geq 2$ (if I am not missing anything!). Indeed, Assumption 3.1 takes into account the expressivity of the activation function. However, note that in your example, one cannot fit $h_*$ anyway (degree $n$ activation and degree $n+1$ activation). More generally, for any $\sigma$ degree $k$ polynomial, I would expect (didn't check) $\sigma (u + v^T z_{S^\perp})$ to span  all polynomials of degree-$k$ and to be equivalent to Assumption (3.2) whenever $h_*$ is degree $\leq k$, which is anyway the only part of the target function that can be fitted by the neural network. Hence, for $\sigma$ degree-$k$ polynomial, we could imagine the following: decompose $h_* = h_{\leq k} + h_{>k}$, assume Assumption (3.2) on $h_{\leq k}$ and disregard $h_{>k}$, which does not contribute to the dynamics by orthogonality. (Note that $h_{\leq k}$ can only have higher leap than $h_*$.) All in all, I am still not convinced that Assumption (3.1) is significantly different from isoleap to claim ``we propose a basis-free generalization of the merged-staircase property in [1]''. I apologize for being nit-picking (and I do not consider it to be the most important part of your paper), but several other groups, including [2,7,9,10] cited in your paper, have worked on this basis-free leap complexity. Please let me know if I misunderstood anything.
>
> - **Concerning Algorithm 1 and Assumption 4.1:** Thank you for the clarification, I mistakenly thought the initialization was a point mass. I acknowledge that the Gaussian case is much harder than the hypercube case, due to the need of an algebraic independence. Thank you for making the effort of verifying Assumption 4.1 in both these cases, I believe it already makes a strong case for Assumption 4.1!

---

> > ### Author Response · Authors · 2024-08-10
> >
> > Dear Reviewer 7m62,
> >
> > Thank you very much for your reply and for the insightful comments. I agree with you that IsoLeap can be defined beyond polynomials and that Equation 3.2 should be equivalent to IsoLeap$\geq 2$. We will add the discussion and clarify this point in our revision. I also agree with you that we probably need to modify the claim of our contribution given the equivalence. But I would still consider the reflective property as interesting -- This property (Equation 3.1) is derived when we compare the original mean-field dynamics (Equation 2.4) and the flow in a subspace (Equation 3.7); after removing the dependence of $\sigma$, a stronger condition (Equation 3.2) equivalent to IsoLeap$\geq 2$ is derived, which re-discovers an interesting condition from a different perspective/analysis. We also give additional insight as Equation 3.1 considers the expressiveness of $\sigma$.

---

> ### Comment · Reviewer_7m62 · 2024-08-12
>
> Thank you for taking into account my comments.
> I increased my score to 5. The setting is interesting and can be of interest to the NeurIPS community. However, I believe that the paper still falls short of solving the main difficulty of going from hypercube to Gaussian, which is the algebraic independence, and requires an unatural construction (repeating step 2).

---

> > ### Author Response · Authors · 2024-08-13
> >
> > Dear Reviewer 7m62,
> >
> > Thank you very much for increasing the score! We appreciate all your valuable comments and will revise our manuscript accordingly.

---

### Official Review · Reviewer_nSKZ · 2024-07-13

**Soundness:** 3
**Presentation:** 3
**Contribution:** 3
**Rating:** 7
**Confidence:** 3

**Summary:**

This paper studies how well two-layer neural networks can learn subspace-sparse polynomials using stochastic gradient descent when the input is Gaussian. It proposes a necessary condition for learning, showing that if the target function has certain symmetries, the neural network cannot learn it perfectly. The paper also provides a sufficient condition and training strategy that guarantee the loss decays exponentially fast to zero, with rates independent of the input dimension.

**Strengths:**

1. This paper is well written, with detailed explanations, proofs, and algorithms.

2. It gives both necessary and sufficient conditions for when neural networks can learn these functions well, which helps understand the limits of what neural networks can do.

3. The results don't depend on the dimension of the input, which means they work even for very high-dimensional data.

**Weaknesses:**

No obvious weakness.

**Questions:**

No questions

---

> ### Author Rebuttal · Authors · 2024-08-07
>
> Thank you very much for your encouraging comments! We appreciate your time and consideration.

---

### Decision · Program_Chairs · 2024-09-25

**Decision:**

Accept (poster)

**Comment:**

The reviewers initially raised several important concerns and the extensive responses of the authors have partially addressed them, which lead the reviewers increase their scores. Given all the material (paper, reviews, rebuttal and discussion), I am recommending an acceptance by trusting the authors that they will implement all the promised changes and clarifications one by one (there are many points to be clarified & elaborated as revealed during the discussion period). Please make sure you go over all the requested changes carefully and update the camera-ready version accordingly.